# Synthesis of Anti-Inflammatory Spirostene-Pyrazole Conjugates by a Consecutive Multicomponent Reaction of Diosgenin with Oxalyl Chloride, Arylalkynes and Hydrazines or Hydrazones

**DOI:** 10.3390/molecules27010162

**Published:** 2021-12-28

**Authors:** Maksim E. Mironov, Sergey A. Borisov, Tatyana V. Rybalova, Dmitry S. Baev, Tatyana G. Tolstikova, Elvira E. Shults

**Affiliations:** 1N.N. Vorozhtsov Novosibirsk Institute of Organic Chemistry, Siberian Branch of the Russian Academy of Sciences, Academician Lavrentyev Ave., 9, 630090 Novosibirsk, Russia; mmironov@nioch.nsc.ru (M.E.M.); sergalborisov@mail.ru (S.A.B.); rybalova@nioch.nsc.ru (T.V.R.); mitja2001@gmail.com (D.S.B.); tg_tolstikova@mail.ru (T.G.T.); 2Department of Natural Sciences, Novosibirsk State University, Piragova Str., 1, 630090 Novosibirsk, Russia

**Keywords:** multi-component synthesis, Stephens–Castro reaction, heterocyclization, diosgenin, pyrazole, anti-inflammatory activity

## Abstract

Steroid sapogenin diosgenin is of significant interest due to its biological activity and synthetic application. A consecutive one-pot reaction of diosgenin, oxalyl chloride, arylacetylenes, and phenylhydrazine give rise to steroidal 1,3,5-trisubstituted pyrazoles (isolated yield 46–60%) when the Stephens–Castro reaction and heterocyclization steps were carried out by heating in benzene. When the cyclization step of alkyndione with phenylhydrazine was performed in 2-methoxyethanol at room temperature, steroidal α,β-alkynyl (*E*)- and (*Z*)-hydrazones were isolated along with 1,3,5-trisubstituted pyrazole and the isomeric 2,3,5-trisubstituted pyrazole. The consecutive reaction of diosgenin, oxalyl chloride, phenylacetylene and benzoic acid hydrazides efficiently forms steroidal 1-benzoyl-5-hydroxy-3-phenylpyrazolines. The structure of new compounds was unambiguously corroborated by comprehensive NMR spectroscopy, mass-spectrometry, and X-ray structure analyses. Performing the heterocyclization step of ynedione with hydrazine monohydrate in 2-methoxyethanol allowed the synthesis of 5-phenyl substituted steroidal pyrazole, which was found to exhibit high anti-inflammatory activity, comparable to that of diclofenac sodium, a commercial pain reliever. It was shown by molecular docking that the new derivatives are incorporated into the binding site of the protein Keap1 Kelch-domain by their alkynylhydrazone or pyrazole substituent with the formation of more non-covalent bonds and have higher affinity than the initial spirostene core.

## 1. Introduction

Diosgenin, (25*R*)-spirost-5-en-3β-ol **1**, is a steroid sapogenin part of the saponin dioscin which can be found in several plant species, including dioscorea, yams and smilax [1,2]. Preclinical studies have implicated the potential use of diosgenin in several ailments, such as cancer, diabetes, hypercholesterolemia, gastrointestinal disorders, and inflammatory conditions. Antioxidant and anti-inflammatory properties [3,4] of diosgenin prevailed in most of the studies which can be attributed to its therapeutic properties on diabetes, cerebral ailments, obesity, skin ageing, cardiovascular diseases, allergic and menopausal syndrome, and also cancer [5,6,7]. The versatile anticancer and anti-inflammatory activity exhibited by diosgenin indicates that this molecule could be a starting point for developing a new medicine, as an alternative drug of natural origin capable of diminishing the side-effects caused by allopathic drugs. However, diosgenin is poorly soluble in physiological media and has low absorption and a high percentage of the absorbed drug is metabolized rapidly [8,9,10]. Therefore, with the aim of improving the biological activity and drugability properties, several diosgenin prodrugs and derivatives were designed and synthesized [11,12,13,14,15,16,17,18,19,20,21,22,23,24]. For obtaining new anti-inflammatory and anticancer agents the modification on the C-3 hydroxy-substituent was in the focus [15,16,17,18,19,20,21,22,23,24]. In the present research a one-pot methodology has been adopted to synthesize novel spirostenol derivatives involving the reaction of diosgenin 1 with oxalyl chloride followed by the Stephens–Castro reaction of 3-*O*-(2-chloro-2-oxoacetyl)-diosgenin **2** with terminal arylacetylenes and heterocyclization of the formed arylalkyne-1,2-diones with phenylhydrazine hydrochloride (Figure 1). Hydrazine monohydrate and benzoic acid hydrazides were also studied in the heterocyclization step for obtaining of spirostene-pyrazole and spirostene-pyrazoline conjugates.

Pyrazoles and pyrazolines are privileged 1,2-diazole derivatives in a broad range of applications, both in life and materials sciences. While the former are fully conjugated and can be considered as heteroaromatic 6π-systems with interesting properties as crop-protecting agents [25], pharmaceutically active ingredients [26,27,28,29,30] and ligands [31], the partially unsaturated 2*H*-pyrazolines have particularly attracted attention in medicinal chemistry, for instance, as antibacterial [32], anti-inflammatory [32,33], anticancer [34,35] and antidepressive [36] agents.

Taking into account the interest in diosgenin and pyrasole derivatives as anti- inflammatory agents, the effect of selected compounds was investigated using a histamine-induced mice paw edema model. The molecular docking of obtained pyrazole-diosgenin conjugates into the binding site of the protein Keap1 Kelch-domain was carried out.

## 2. Results and Discussion

### 2.1. Chemistry

Alkyne-1,2-diones are of interest for the synthesis of a variety of functionalized and heterocyclic compounds because, as densely functional trielectrophiles (the alkyne, alkynone and dicarbonyl function), they can be easily and selectively transformed by a series of nucleophilic additions, annulations, condensations, and Diels–Alder reactions [37,38,39,40,41,42,43]. It is highly desirable to develop efficient reaction systems for the synthesis of functionalized pyrazoles from alkyne-1,2-diones and arylhydrazines with high efficiency and regioselectivity under mild reaction conditions. Described atom- and step-economic synthesis of these heterocycles included the sequence of glyoxylation-alkynylation of aryl(hetaryl) substrates [38,44] or activation-alkynylation of (hetero)arylglyoxylic acids and the heterocyclization reaction of the resulting alkynyl diketones [39,40].

Initially, we worked out a three-component synthesis of alkyne-1,2-diones of diosgenin **1**. Diosgenin **1** was first converted to the steroidal monooxalyl chloride 2 by treating with an excess of oxalyl chloride in chloroform at 0 °C. After removing the excess of oxalyl chloride and chloroform in vacuo, the resulting 3β-*O-*(2-chloro-2-oxo-aceta- te)-5-spirostene **2** (Figure 2) was involved in the Stephens–Castro cross-coupling reaction with aryl alkynes (**3a**–**e**). It was known that cross-coupling of terminal acetylenes with monooxalyl chloride giving acetylenic diketones is best performed in ethereal solvent in the presence of CuI (5% mol) and Et_3_N (2 mmol) in ligand-free conditions. The Stephens–Castro reaction of steroidal oxalyl chloride **2** and phenylacetylene **3a** in THF in the mentioned conditions led to compound **4** in a yield of 9%. Varying amounts of copper iodide (5, 10, 20 mol %) and Et_3_N (2 or 3 equiv.) and also using the additive (TMEDA, 0.1 equiv.) gave no acceptable results; the alkyne-1,2-dione **4** was obtained in trace amounts. Further, it was not possible to obtain the target compound **4** when replacing the base with N,N-diisopropylethylamine (DIPEA) or K_2_CO_3_, and the solvent with 1,4-dioxane. The best result for the coupling step was obtained by replacing ether solvents with benzene, reducing the amount of Et_3_N to 1 equiv. and increasing the amount of catalyst to 10 mol %. By performing the reaction at 40 °C, compound **4** was isolated in 48% yield after column chromatography. Increasing the reaction temperature diminishes the yield, and prolonging the reaction time (more than 24 h) does not increase the yield. Under the found conditions, spirostene ynediones **5**–**8**, containing fragments of aryl acetylenes with 4-methoxy-, 4-fluoro-, 4-ethyl- and 3-methyl substituent, were obtained in 18–48% isolated yields. Characteristically, the content of alkyne-1,2-diones **4**–**8** in the reaction mixture was 70–80% according to ^1^H NMR data; however, a significant part of the products was decomposed during isolation by chromatography on silica gel due to the low stability of alkyne-1,2-diones. The structure of (22*R*,25*R*)-spirost-5-en-3β-yl 2-oxo- 4-(phenyl)but-3-ynoate **4** has been corroborated by an X-ray structure analysis (Figure 1).

Inspired by the alkynylation of in situ generated 3β-*O-*(2-chloro-2-oxoacetate)-5-spirostene **2** and taking into account the low stability of compounds **4**–**8**, we set out to design a one-pot alkynylation-heterocyclization sequences that start from diosgenin **1** and apply treatment with oxalyl chloride for formation of 3β-*O-*(2-chloro-2-oxoacetate)-doiosgenin, Stephens-Castro conditions for the generation of ynediones **4**–**8** and their reaction with phenylhydrazine (Figure 3). The stage of heterocyclization was carried out by adding phenyl hydrazine hydrochloride **9** (1 equiv.) to the reaction mixture containing steroidal alkyne-1,2-diones **4**–**8** and triethylamine (1 equiv.) in benzene. The heterocyclization step, similar to the Stephens–Castro cross-coupling reaction, proceeded in benzene but required an increase in temperature to 60 °C. After 24 h (indicated by TLC), the corresponding 5-aryl substituted 3-*O*-(pyrazol-3-yloxo)diosgenin derivatives **10**–**14** were obtained with an isolated yield of 46–60% (Figure 3). The structures of 3-*O*-(pyrazolocarbonyl)spirostenes **11**,**12** were confirmed by X-ray structural analysis (Figure 2). Importantly, excellent regioselectivity was observed, and only one regioisomer was formed during this one-vessel process.

Annulation reactions of ynediones have been broadly explored, and these reactions usually need additional catalyst [43,45,46] or high temperature [39]. In our case, the copper(I)/triethylamine-catalyzed synthesis of pyrazoles in yields up to 46–60% was accomplished in one vessel from the reaction of diosgenin **1** with oxalyl chloride at room temperature, with the removal of the solvent from the reaction mixture, diluting with benzene and sequential addition of triethylamine, CuI, aryl alkynes **3a**–**e**, and phenylhydrazine hydrochloride **9**.

Based on the analysis of the spectral and analytical data of obtained pyrazoles **10**–**14** we can suggests that the heterocyclization reaction of alkyne-1,2-diones **4**–**8** started with the nucleophilic addition of phenylhydrazine at the carbonyl group conjugated with the triple bond to form the corresponding α,β-alkynyl phenylhydrazones, which then undergo intramolecular cyclization. Heterocyclization of (hetero)aryl alkyne-1,2-diones with N-acylhydrazines with the formation of pyrazole derivatives was performed in a medium of protic solvents—methanol or 2-methoxyethanol [38,40]. We also attempted to use ethanol at the stage of heterocyclization of spirostane alkyne-1,2-diones **4**–**8** with phenylhydrazine hydrochloride; however, their low solubility did not allow for achieving the formation of the target steroidal pyrazoles. At the same time, when using a mixture of ethanol and benzene as a reaction medium, it was possible to isolate as reaction products 1,3,5-trisubstituted pyrazoles **10**–**14** (yield 9–69%) and steroidal α,β-alkynyl-substituted (*E*)-hydrazones **15**–**19** (yields 27–37%) (Figure 4, Table 1). The use of 2-methoxyethanol as a solvent in the reaction of spirostene alkyne-1,2-dione **4** with phenylhydrazine hydrochloride **9** (conditions *c*) led to a significant increase in the yield of α,β-alkynyl-substituted *(E*)-hydrazone **15** to 47%; the α,β-alkynyl-substituted (*Z*)-hydrazone **20** (yield 7%) and 1,3,5-trisubstituted pyrazoles 10 (yield 12%) were also isolated by column chromatography. Additionally, the isomeric 2,3,5-trisubstituted pyrazole **21** (yield 7%)—the product of the Michael addition/enamine-hydrazone tautomerization—and heterocyclization pathway were isolated (Figure 4).

It is noteworthy that in mild reaction conditions by using of 2-methoxyethanol as the solvent we were able to isolate alkynyl substituted steroidal (*E*)-hydrazones **15**–**19**, in several examples, as the main products. We found that (*Z*)-hydrazone **20** has been not so stable in the reaction conditions; on heating to 40–60 °C in benzene, or on silica gel during chromatography, this compound partially converted into the stable (*E*)-izomer **15**. The structures of steroidal (*E*)- and (*Z*)-phenylhydrazones **15**,**20** and minor 2,3,5-trisubstituted pyrazole **21** were confirmed by the data of X-ray structural analysis (Figure 3). 

It must been noted that α,β-alkynic hydrazones are an important class with practical value, they have been emanated as powerful synthons for constructing diverge ranges of cyclic compounds through transition metal catalyzed or transition metal free reactions [47,48,49]. 

The sequential four-component reaction of diosgenin 1, oxalyl chloride, phenylacetylene 3a, and hydrazine monohydrate **22** occurred smoothly and led to the formation of 3,5-disubstituted pyrazole **23** (Figure 5, Figure 4). The final step of this sequence consisted of Michael addition/cyclocondenzation/elimination reactions, and ran in 2-methoxyethanol at room temperature.

N-Acyl substituted hydrazones, as expected, showed a softer character of the nucleophilic center, which is expressed in their initial addition at the soft electrophilic center of the triple bond of alkynyl ketones with the formation of intermediate alken-hydrazide A and their subsequent intramolecular cyclization leading to 1-hydroxy-4,5-dihydro-1*H*-pyrazoles [37]. The reactions of the in situ formed alkynyldiketones with N-acylhydrazides usually need an additional catalyst or high temperatures [40]. We found, that the consecutive four-component reaction of diosgenin **1**, oxalyl chloride, phenylacetylene **3a**, and hydrazides of benzoic or 4-bromobenzoic acids **24a**,**b** led to the formation of 3-*O*-(1-aroyl-5-hydroxy-3-phenyl-4,5-dihydro-1*H*-pyrazoline)- spirostenes **25** or **26** isolated in the yield of 49 and 42%, respectively (Figure 6). Thus, the initial reaction of N-acylhydrazines with the in situ formed steroidal alkynyldione proceeds through the addition of the nucleophile at the triple bond and the formation of intermediate alken-hydrazide A. Additionally, in the reaction of **1** with 4-bromobenzoic acid hydrazide **24b**, the aromatizing elimination of water and simultaneous deacylation of 4-bromophenacyl substituent furnishing of compound **23** (Figure 4) was observed (yield 18%).

The structures of synthesized compounds were established by ^1^ H NMR and ^13^ C, and IR spectroscopies, mass spectrometry, and elemental analysis data. The ^1^H NMR spectrum of alkynyl-substituted (*E*)-hydrazones 15-19 in deuterochloroform is characterized by the presence of an NH-proton signal in the form of a broad singlet at 9.05–9.09 ppm, while the signal of a similar proton of (Z)- hydrazone 20 appears as a singlet at δ12.80 ppm. Two alkynic carbons (C-3′ and C-4′), in the spectra of alkynyl substituted (Z)-hydrazone 20 resonate closely at δ 85.3 and 90.0 ppm (Δδ 4.7 ppm). In the (*E*)-isomers **15**–**19**, the alkynic carbon adjacent to the carbonyl group is comparatively upfield (δ 77.3–77.8 ppm), while the other alkynic carbon (C-4′) is relatively downfield (δ 103.6–105.2 ppm) and the chemical shift difference between these carbons is roughly 26.1−28.3 ppm. In brief, the absolute value of chemical shift difference between alkynic carbons in the (*E*)-isomer is typically bigger as compared to that in the (*Z*)-isomer. A characteristic down-field shift was observed for the C-2′ carbon atom (δ 117.6–118.2 ppm) for α,β-alkynyl-substituted (*E*)-hydrazones 15-19 as compared to the signal for (*Z*)-isomer (δ 113.9 ppm). The diastereotopic protons H-4′ in the ^1^H NMR spectra of pyrazolines **25** and **26** manifested as doublets at δ 3.40, 3.60 (**25**) and 3.43, 3.60 ppm (**26**) (*J* = 17.9 Hz), respectively. The formation of a mixture of diastereomeric 5-hydroxypyrazolines is indicated by the doubling of some signals in the ^13^C NMR spectrum; the largest difference was observed for the chemical shifts of carbon atom C-5′ (δ 89.5; 89.6 ppm) (Appendix A, ^1^H and ^13^C NMR spectra of compounds **25** and **26**; pp. 39–42).

The structure of compounds **4**, **11**, **12**, **15**, **20**, **21** and **23** were determined by single crystal X-ray analysis (Figure 1, Figure 2, Figure 3 and Figure 4). The analysis of the molecular geometries was performed using the PLATON program [50,51]. All the compounds contain the spirosten six-ring moiety: the tetrahydropyran cycle is spiro fused with the furan cycle, which is *cis*-fused with the cyclopentan ring, and the rest are *trans*-fused in all the compounds observed. The cyclohexane and pyran cycles adopt a chair conformation, the cyclohexene—a halfchair conformation. The furan cycle has an envelope conformation with the deviation of the O1 atom from the rest of the atoms of the cycle in **4**, **12**, and C22 in **15**. The different deviating atom in the furan cycle of **15** causes the absence of the intramolecular hydrogen bond C26-H26A…O1 occurred in **4**, **11** and **12**, with parameters shown in Table 2. The cyclopentane ring has a twist conformation in **4**, **11, 12**, and an envelope in **15**. The intramolecular hydrogen bond C3-H3…O4 keeps atom O4 almost in the plane of C3O3C1′ in all compounds.

In molecule **4** the deviations of C3′, C4′ and C1″ from the plain (within ± 0.089(3) Å) of the C3′C2′(=O5)C1′(=O4)O3) moiety increases from C3′ to C1″ equaling to 0.097(3), 0.265(3) and 0.519(3) Å accordingly and inter-plain angle of this moiety with phenyl ring C1″ ÷ C6″ equals to 26.3(7)^o^. 

Compound **15** with a (*E*)-configuration of N1=C2′ and O3C1′(=O4)C2′=(N1)C3′C4′C1″ moiety is plain within ± 0.0173(3) Å, with the deviations of N2 and C1‴ atoms from this plain equaling to 0.045(3) and 0.278(3) Å accordingly, and inter-plain angles to phenyl rings C1″-C6″ and C1‴-C6‴ being equal to 17.24(6) and 12.73(6) ^o^.

An isomer of **15,** the compound **20**, having (*Z*)-configuration of N1′=C2′, is characterized with existing of strong intramolecular hydrogen bond N2′-H2′…O4 (Table 1), that leads to an almost plane substituent in C3. Inter-plane angles of the hydrogen bond cycle O4=C1′C2′=N1′N2′-H2′ with phenyl rings C1″ ÷ C6″ and C1‴ ÷ C6‴ are 6.85 and 10.74 accordingly; that is less that in **15**. The bond lengths distribution of O4C1′C2′=N1′N2′C1‴ moiety is also different for **15** and **20** (Table 3).

The common feature of **11**, **12**, **21** and **23** structures is that C1′(=O4)O3 moiety lays practicaly in the pyrazol plane that can be convenient to form common conjugated π-system. Thus, the torsion angle N1′C5′C1′O3 is −9.9(4)^o^ (**11**), −11.2(3)^o^ (**12**), −9.8(4) and −1.0(5)^o^ for two independent molecules of (**21**) and 0.7(5)^o^ (**23**). 

As the crystals of compounds **11** and **12** are isomorphous, their geometrical parameters are very close. The inter-plane angles of pyrazol with phenyls C1″-C6″ and C1‴-C6‴ are: 41° (**12**), 37° (**11**) for first one, and 64° (**12**), 61° (**11)** for the second. As for **21**, such angles are 13 and 9° for C1″-C6″ and 70 and 63° for C1‴-C6‴ in two independent molecules. The same angle between the pyrazol ring and phenyl substituent C1″ ÷ C6″ in **23** equals 25°.

### 2.2. Anti-Inflammatory Activity of Diosgenin 1 and Its 3-O-Substituted Derivatives in the Histamine-Induced Paw Edema Model

As previously mentioned, diosgenin **1** is a multi-targeted agent that has immense potential to be used as a wonder drug for the treatment of innumerable chronic diseases, such as cancer, CVDs, metabolic and nervous system disorders, and different types of inflammatory diseases [52,53]. Diosgenin **1** and its 3-*O*-substituted derivatives **7**,**8**,**10**,**13**–**16**,**18**,**19**,**21**,**23** and **26** were tested for their anti-inflammatory activities using the histamine-induced paw edema model [54]. The results are presented in Table 4.

Data in Table 4 illustrate that diosgenin **1**, alkynyldiketones **7**,**8**, hydrazones **15**–**19**, and pyrazole conjugate **23**, administered orally in mice (50 mg/kg dose), decreased the paw edema by 29–59 percent in relation to the control group. The same attribute of the reference drug diclofenac sodium in the effective dose of 10 mg/kg was 56 percent. Diosgenin-pyrazole conjugate **23** showed the highest activity. However, substituent on the nitrogen atoms in the pyrazole ring led to a decrease in anti-inflammatory activity (**10**, **13**, **14**, **21**, **26**). The activity of alkynyldiketones **7**,**8**, hydrazones **15**,**18** and 3,5-disubstituted pyrazole derivative **23** compares favorably with that of the starting compound **1**. These data indicate that the anti-inflammatory activity of diosgenin derivatives is sensitive to the nature of substitution at the 3-hydroxy substituent and confirms the potential value of this class of biological active substances. 

### 2.3. Molecular Docking

The literature search resulted in many in vitro, in vivo and clinical trials that reported the efficacy of diosgenin and its analogs in modulating important molecular targets and signaling pathways, such as PI3K/AKT/mTOR, JAK/STAT, NF-κB, and NFkBp65, which play a crucial role in the development of most of the chronic diseases [53]. It can modulate antioxidant defense and decrease oxidative stress damage [55] and exhibit inhibitory effects on superoxide anion production through the blockade of cAMP, PKA, cPLA2, PAK, Akt and MAPKs signaling pathways [56]. There are three main cellular components involved in the regulation of antioxidant response, and they are Kelch-like ECH-associated protein 1 (Keap1), nuclear factor erythroid 2-related factor 2 (Nrf2), and antioxidant response elements (ARE). It is now intensively studied that activation of Keap1-Nrf2-ARE signaling can provide protection against various stress and inflammation related diseases, including neurogenerative diseases, autoimmune diseases, and cardiovascular disorders [57]. From the molecular docking assay results, the parent saponin dioscin (glycoside form of diosgenin) showed powerful affinities towards to Sirt1, Keap1 and NF-κBp65, indicating that the compound may directly bind to these proteins, to exert its biological activities [58]. In this study we aimed to provide in silico evidence that pyrazole–diosgenin hybrids bind Keap1 and hence could be employed as promising Nrf2 activators. AutoDock Vina 1.5.6 was used to perform molecular docking of diosgenin **1**, and its derivatives **15** and **23**.

The tunnel-like binding site of the Kelch domain is substantially hydrophilic. It has a fairly limited size. We can also observe that large hydrophobic diosgenin scaffold **1** (−3.169 kcal/mol) cannot penetrate deep into the binding site (Figure 5A). However, the introduction of a substituent at the C-3 position of the diosgenin structure significantly reduces the estimated minimal binding energy of the steroidal α,β-alkynyl (*E*)-hydrazone **15** (−6.344 kcal/mol) and pyrazole **23** (−4.828 kcal/mol) to the Kelch domain, which is probably associated with the ability of the substituents to interact with amino acids deep in the binding site (Figure 5B,C). The superposition of structures in the binding site indicates that the diosgenin scaffolds of the studied molecules remain in the wide outer part of the binding site. Functionalization of the hydroxyl group of the diosgenin ring A leads to a very close orientation of the diosgenin scaffold of derivatives in the space of the binding site (Figure 5D). The structures of the substituents of the new diosgenin derivatives are stabilized in the deep part of the binding site due to hydrophobic interactions. Attention is drawn to the formation of a hydrogen bond of the same type for compounds **15** and **23** with an amino acid residue Ser602. The hydroxyl group of diosgenin **1** can probably form a hydrogen bond with the amino acid residue Ser508.

## 3. Materials and Methods

### 3.1. Chemistry

#### 3.1.1. General Information

^1^H and ^13^C NMR spectra were acquired on ‘Bruker AV 300′ 300.13 (^1^H) and 75.47 MHz (^13^C), respectively, (compounds **14**, **15**), ‘Bruker AV 400′ 400.13 (^1^H) and 101.61 MHz (^13^C) (**2**, **5**, **6**, **13**, **18**), ‘Bruker DRX-500′ 500.13 (^1^H) and 125.76 MHz (^13^C), (**19**) and ‘Bruker AV 600′, 600.30 (^1^H) and 150.94 MHz (^13^C), (**4**, **10**, **16**, **17**, **21**, **23**, **25**, **26**) instruments. The ^1^H spectra for compounds **7**, **11**, **12** were obtained on ‘Bruker AV 400′, for **8, 2**—on ‘Bruker AV 300′, for **21**—on ‘Bruker AV 600′, ^13^C spectra for **7**, **8**, **11**, **12**, **20**, **21**—on ‘Bruker DRX-500′. Deuterochloroform (CDCl_3_) was used as a solvent, with residual CHCl_3_ (δ_H_ = 7.24 ppm) or CDCl_3_ (δ_C_ = 77.0 ppm) being employed as internal standards. NMR signal assignments were carried out with the aid of a combination of 1D and 2D NMR techniques that included ^1^H, ^13^C, COSY, HSQC, and HMBC spectra. IR absorption spectra were recorded on a Vector 22 FT-IR spectrometer in KBr pellets. The specific rotation values [α]_D_ were obtained on a PolAAr 3005 polarimeter. Melting points were determined using termosystem Mettler Toledo FP900 (Columbus, OH, USA). HRMS spectra were recorded on a DFS mass spectrometer (Thermo Fisher Scientific, Waltham, MA, USA), evaporator temperature 180–220 °C, EI ionization at 70 eV). Elemental analysis was carried out on a 1106 Elemental analysis instrument (Carlo-Erba, Milan, Italy). The X-ray diffraction experiments for crystals of **4**, **11**, **12**, **15**, **20**, **21** and **23** were performed at ambient conditions on a Bruker KAPPA APEX II diffractometer (graphite-monochromated Mo Kα radiation). Reflection intensities were corrected for absorption by SADABS program [59]. The structure was solved by direct methods using the SHELXS-97 program and refined by an anisotropic (isotropic for all H atoms) full-matrix least-squares method against *F^2^* of all reflections by SHELX-97 [60]. The positions of the hydrogen were calculated geometrically and refined in a riding model. 

The reaction progress and the purity of the obtained compounds were monitored by TLC on Silufol UV-254 plates (CHCl_3_-EtOH, 9:1; detection under UV light or by treatment with iodine vapor). The reaction progress and the purity of the obtained compounds were monitored by TLC on Silufol UV-254 plates (Kavalier, Czech Republic, CHCl_3_-EtOH, 100:1; detection under UV light or by spraying the plates with a 10% water solution of H_2_SO_4_ followed by heating at 100 °C). Products were isolated by column chromatography on silica gel 60 (0.063–0.200 mm, Merck KGaA, Darmstadt, Germany), eluting with indicated solvent systems. The chemicals used: arylacetylenes **3a**–**e**, oxalyl chloride, phenyl hydrazine hydrochloride **9**, hydrazine monohydrate **22**, benzhydrazide **24a** and 4-bromobenzhydrazide **24b**, were purchased from Aldrich (St. Louis, MO, USA) or Alfa Aesar (GmbH, Karlsruhe, Germany). Solvents (CHCl_3_, 2-methoxyethanol, ethanol, benzene) and Et_3_N were purified by standard methods and distilled under a stream of argon just before use. Copies of NMR spectra (^1^H and ^13^C) are provided in Appendix A. 

#### 3.1.2. Synthesis and Spectral Data

##### The Acylation Reaction of Diosgenin 1 with Oxalyl Chloride

A solution of diosgenin **1** (500 mg, 1.21 mmol) in CHCl_3_ (10 mL) was added dropwise to a cold stirred solution of oxalyl chloride (621 mg (0.42 mL, 4.9 mmol) in CHCl_3_ (3 mL) in argon atmosphere at 0 °C for 1 h. The reaction mixture was stirred at room temperature for 3 h. The solvent was removed under reduced pressure, the residue was treated with CHCl_3_ (3mL) and additionally evaporated. This procedure was repeated three times for removing the trace of oxalyl chloride. As result, light yellow oil of compound **2** was obtained and used without purification in a Stephens–Castro reaction with terminal arylacetylenes. [(22*R*,25*R*)-S*pirost-5-en-3β-yl 2-chloro-2-oxoacetate, {**3-O-(2-chloro-2-oxoacetyl)diosgenin}* (**2**). ^1^H NMR (400 MHz, CDCl_3_, δ, ppm): 0.75 (3H, d, *J* = 6.8 Hz, H-27), 0.76 (3H, s, H-19), 0.95 (3H, d, *J* = 7.0 Hz, H-21), 0.98 (1H, m, H-9), 1.05 (3H, s, H-18), 1.09–1.23 (3H, m, H-1,12,14), 1.30 (1H, td, *J* = 12.1, 6.5 Hz, H-15), 1.41–2.04 (17H, m, H-1,2,2,7,7,8,11,11,12,15,17,20,23,23, 24,24,25), 2.40–2.51 (2H, m, H-4), 3.36 (1H, t, *J* = 10.9 Hz, H-26), 3.45 (1H, dd, *J* = 10.9, 4.0 Hz, H-26), 4.40 (1H, dd, *J* = 14.9, 7.5 Hz, H-16), 4.77 (1H, m, H-3), 5.41 (1H, dd, *J* = 3.4, 2.2 Hz, H-6). ^13^C NMR (101 MHz, CDCl_3_, δ, ppm): 14.5 (C-21), 16.3 (C-18), 17.1 (C-27), 19.2 (C-19), 20.8 (C-11), 27.3 (C-2), 28.8 (C-24), 30.2 (C-25), 31.3 (C-8), 31.4 (C-23), 31.8 (C-15), 32.0 (C-7), 36.6 (C-10), 36.7 (C-1), 37.4 (C-4), 39.6 (C-12), 40.2 (C-13), 41.6 (C-20), 49.8 (C-9), 56.3 (C-14), 61.9 (C-17), 66.8 (C-26), 79.5 (C-3), 80.8 (C-16), 109.3 (C-22), 123.6 (C-6), 138.5 (C-5), 155.1 (C-1′), 161.4 (C-2′). 

##### One-Pot Synthesis of Steroidal Ynediones (**4**–**8**)

A solution of diosgenin (800 mg, 1.93 mmol) in CHCl_3_ (10 mL) was added dropwise to a cold stirred solution of oxalyl chloride (980 mg (0.66 mL), 7.72 mmol) in CHCl_3_ (5 mL) in argon atmosphere at 0 °C for 1 h and the reaction mixture was stirred at room temperature for 3 h. The solvent was removed under reduced pressure, the residue was treated with CHCl_3_ (3mL) and additionally evaporated. This procedure was repeated three times for removing the trace of oxalyl chloride and diluted with C_6_H_6_ (10 mL) in an argon flow. The corresponding aryl acetylene **3a**–**e** (1.00 mmol), CuI (19 mg, 0.1 mmol) and Et_3_N (0.138 mL, 1.00 mmol) were added subsequently in an argon flow at room temperature. The mixture was heated under stirring at 40 °C for 12 h (TLC), after that solvent was removed under reduced pressure, and the residue was purified by column chromatography (eluent petroleum ether–ether, 20:1) to give compounds (**4**–**8**).

**Compound 4**. (4*S*,5′*R*,6a*R*,6b*S*,8a*S*,8b*R*,9*S*,10*R*,11a*S*,12aS,12b*S*)-5′,6a,8a,9-Tetramet- hyl-1,3,3′,4,4′,5,5′,6,6a,6b,6′,7,8,8a,8b,9,11a,12,12a,12b-icosahydrospiro[naphtho [2′,1′:4,5]- indeno [2,1-*b*]furan-10,2′-pyran]-4-yl 2-oxo-4-phenylbut-3-ynoate [(22*R*,25*R*)-spirost-5-en -3β-yl 2-oxo-4-phenylbut-3-ynoate] (**4**). Yield 48%. Yellowish needles. M.p. 209–212 °C (decomp.) (petroleum ether–diethyl ether, 20:1). [α]_D_^2^^7^—88.5 (c 0.4, CHCl_3_). ^1^H NMR (600 MHz, CDCl_3_, δ, ppm): 0.74 (3H, d, *J* = 6.8 Hz, H-27), 0.77 (3H, s, H-19), 0.98 (3H, d, *J* = 7.0 Hz, H-21), 1.01 (1H, m, H-9), 1.08 (3H, s, H-18), 1.10–1.22 (3H, m, H-1,12,14), 1.25–1.33 м (1H, m, H-15), 1.44–1.50 (2H, m, H-11,24), 1.54 (1H, m, H-11), 1.58–1.69 (6H, m, H-7,8,23,23,24, 25), 1.75 (1H, dm, *J* = 8.9 Hz, H-12), 1.79 (2H, m, H-2,17), 1.88 (1H, m, H-20), 1.93 (1H, dt, *J* = 13.5, 3.3 Hz, H-1), 1.98–2.06 (3H, m, H-2,7,15), 2.46 (1H, ddd, *J* = 13.3, 4.9, 1.9 Hz, H-4), 2.52 (1H, td, *J* = 11.3, 2.0 Hz, H-4), 3.38 (1H, t, *J* = 11.0 Hz, H-26), 3.48 (1H, dd, *J* = 11.0, 2.3 Hz, H-26), 4.42 (1H, dd, *J* = 15.2, 7.4 Hz, H-16), 4.82 (1H, m, H-3), 5.43 (1H, dd, *J* = 3.5, 1.9 Hz, H-6). 7.43 (2H, dd, *J* = 8.2, 7.8 Hz, H-3″,5″), 7.53 (1H, t, *J* = 7.8 Hz, H-4″), 7.67 (2H, d, *J* = 8.2 Hz, H-2″,6″). ^13^C NMR (151 MHz, CDCl_3_, δ, ppm): 14.5 (C-21), 16.2 (C-18), 17.1 (C-27), 19.3 (C-19), 20.8 (C-11), 27.4 (C-2), 28.8 (C-24), 30.3 (C-25), 31.4 (C-8), 31.4 (C-23), 31.8 (C-15), 32.1 (C-7), 36.7 (C-10), 36.9 (C-1), 37.7 (C-4), 39.7 (C-12), 40.3 (C-13), 41.6 (C-20), 49.9 (C-9), 56.4 (C-14), 62.2 (C-17), 66.8 (C-26), 77.4 (C-3), 80.8 (C-16), 87.2 (C-3′), 97.8 (C-4′), 109.3 (C-22), 119.2 (C-1″), 123.1 (C-6), 128.8 (C-3″,5″), 131.7 (C-4″), 133.8 (C-2″,6″), 139.0 (C-5), 158.8 (C-1′), 169.9 (C-2′). IR (KBr, ν, cm-1): 2195 (C≡C), 1740, 1680(C=O), 1066, 1051, 1024, 1007 (C-O-C), 1597, 814, 760, 688 (C=C). HR-MS, *m*/*z* (I_rel_., %): 570 (0.2), 397 (21), 396 (42), 321 (20), 139 (34), 129 (100), 105 (22), 97 (18), 93 (18), 69 (23), 55.0 (32), 41 (20). Calcd. C_37_H_46_O_5_, *m*/*z* [M]+ 570.3340. Found, *m*/*z*: 570.3339. X-ray structural analysis of compound 4: C_37_H_46_O_5_, M 570.74, monoclinic, P2_1_, a 11.4807(5), b 7.2087(2), c 19.2482(7) Å, β 93.359(2)° V 1590.3(1) Å^3^, Z 2, D_calcd_ 1.192 g·cm^−3^, μ(Mo-Kα) 0.078 mm^−1^, F(000) 616, (θ 1.66–25.05°, completeness 99.9%), colorless, (1.00 × 0.77 × 0.15) mm^3^, transmission 0.8010–0.8620. The intensities of 5632 independent reflections were measured (R_int_ 0.0401), 383 parameters, R_1_ 0.0473 (for 4473 observed I > 2σ(I)). The final refinement parameters: wR_2_ 0.1360, GOOF 1.065, largest diff. peak and hole 0.469 and −0.240 e.A^−3^. Crystallographic data for structure **4** have been deposited at the Cambridge Crystallographic Data Centre as supplementary publication no. CCDC 2077781.

**Compound 5**. (4*S*,5′*R*,6a*R*,6b*S*,8a*S*,8b*R*,9*S*,10*R*,11a*S*,12a*S*,12b*S*)-5′,6a,8a,9-Tetramet- hyl-1,3,3′,4,4′,5,5′,6,6a,6b,6′,7,8,8a,8b,9,11a,12,12a,12b-icosahydrospiro[naphtho[2′,1′:4,5]indeno[2,1-*b*]furan-10,2′-pyran]-4-yl 4-(4-ethylphenyl)-2-oxobut-3-ynoate [(22*R*,25*R*)-spirost-5-en-3β-yl 4-(4-ethylphenyl)-2-oxobut-3-ynoate] (**5**). Yield 35%. Yellowish solid. M.p. 79 °C (decomp.). [α]_D_^2^^5^—72.6 (c 0.2, CHCl_3_). ^1^H NMR (400 MHz, CDCl_3_, δ, ppm): 0.74 (3H, d, *J* = 6.8 Hz, H-27), 0.77 (3H, s, H-19), 0.98 (3H, d, *J* = 7.0 Hz, H-21), 1.01 (1H, m, H-9), 1.08 (3H, s, H-18), 1.10–1.34 (7H, m, H-1,12,14,15, C^8″^H_3_), 1.40–2.05 (17H, m, H-1,2,2,7,7,8,11,11,12,15,17,20,23,23,24,24,25), 2.49 (2H, m, H-4), 2.70 (2H, q, *J* = 7.5 Hz, H-7″), 3.38 (1H, t, *J* = 10.8, H-26), 3.47 (1H, dm, *J* = 10.8 Hz, H-26), 4.41 (1H, dd, *J* = 14.9, 7.5 Hz, H-16), 4.81 (1H, m, H-3), 5.43 (1H, d, *J* = 5.0 Hz, H-6), 7.25 (2H, d, *J* = 8.2 Hz, H-3″,5″), 7.59 (2H, d, *J* = 8.2 Hz, H-2″,6″). ^13^C NMR (101 MHz, CDCl_3_, δ, ppm): 14.5 (C-21), 15.0 (C-8″), 16.3 (C-18), 17.1 (C-27), 19.3 (C-19), 20.8 (C-11), 27.4 (C-2), 28.8 (C-24), 29.1 (C-7″), 30.2 (C-25), 31.3 (C-8), 31.4 (C-23), 31.8 (C-15), 32.0 (C-7), 36.7 (C-10), 36.8 (C-1), 37.6 (C-4), 39.6 (C-12), 40.2 (C-13), 41.6 (C-20), 49.8 (C-9), 56.4 (C-14), 62.0 (C-17), 66.8 (C-26), 77.3 (C-3), 80.8 (C-16), 87.3 (C-3′), 98.9 (C-4′), 109.3 (C-22), 116.2 (C-1″), 123.1 (C-6), 128.4 (C-3″,5″), 134.0 (C-2″,6″), 138.9 (C-5), 149.0 (C-4″), 158.8 (C-1′), 169.9 (C-2′). IR (KBr, ν, cm^−1^): 2197 (C≡C), 1736, 1676 (C=O), 1080, 1025, 1007 (C-O-C), 1605, 1508, 793, 756, 717 (C=C). HR-MS, *m*/*z* (I_rel__._, %): 598 (1), 397 (37), 396 (67), 321 (28), 285 (36), 282 (100), 253 (21), 139 (90), 157 (47), 106 (39). Calcd. C_39_H_50_O_5_. *m*/*z* [M]^+^ 598.3653. Found, *m*/*z*: 598.3652.

**Compound 6**. (4*S*,5′*R*,6a*R*,6b*S*,8a*S*,8b*R*,9*S*,10*R*,11a*S*,12a*S*,12b*S*)-5′,6a,8a,9-Tetramet- hyl-1,3,3′,4,4′,5,5′,6,6a,6b,6′,7,8,8a,8b,9,11a,12,12a,12b-icosahydrospiro(naphtho[2′,1′:4,5]indeno[2,1-*b*]furan-10,2′-pyran)-4-yl 4-(4-methoxyphenyl)-2-oxobut-3-ynoate [(22*R*,25*R*)-spirost-5-en-3β-yl 4-(4-methoxy-phenyl)-2-oxobut-3-ynoate] (**6**). Yield 18%. Yellowish solid. M.p. 111 °C (decomp.). [α]_D_^25^ -92.9 (c 0.2, CHCl_3_). ^1^H NMR (400 MHz, CDCl_3_, δ, ppm): 0.74 (3H, d, *J* = 6.8 Hz, H-27), 0.77 (3H, s, H-19), 0.98 (3H, d, *J* = 7.0 Hz, H-21), 1.01 (1H, m, H-9), 1.08 (3H, s, H-18), 1.12–1.34 (4H, m, H-1,12,14,15), 1.41–1.96 (14H, m, H-1,2,7,8,11,11,12,17,20,23,23,24,24,25), 1.97–2.05 (3H, m, H-2,7,15), 2.49 (2H, m, H-4), 3.38 (1H, t, *J* = 10.8 Hz, H-26), 3.48 (1H, dm, *J* = 10.8 Hz, H-26), 3.87 (3H, s, H-OCH_3_), 4.41 (1H, dd, *J* = 14.5, 7.5 Hz, H-16), 4.81 (1H, m, H-3), 5.43 (1H, d, *J* = 5.0 Hz, H-6), 6.92 (2H, d, *J* = 8.6 Hz, H-3″,5″), 7.63 (2H, d, *J* = 8.6 Hz, H-2″,6″). ^13^C NMR (101 MHz, CDCl_3_, δ, ppm): 14.5 (C-21), 16.3 (C-18), 17.1 (C-27), 19.3 (C-19), 20.8 (C-11), 27.4 (C-2), 28.8 (C-24), 30.3 (C-25), 31.3 (C-8), 31.3 (C-23), 31.8 (C-15), 32.0 (C-7), 36.7 (C-10), 36.8 (C-1), 37.6 (C-4), 39.7 (C-12), 40.2 (C-13), 41.6 (C-20), 49.8 (C-9), 55.5 (OCH_3_), 56.4 (C-14), 62.0 (C-17), 66.8 (C-26), 77.2 (C-3), 80.8 (C-16), 87.8 (C-3′), 99.8 (C-4′), 109.3 (C-22), 110.8 (C-1″), 114.6 C-3″,5″), 123.1 (C-6), 136.1 (C-2″,6″), 139.0 (C-5), 158.9 (C-1′), 162.6 (C-4″), 169.7 (C-2′). IR (KBr, ν, cm^−1^): 2189 (C≡C), 1736, 1668 (C=O), 1080, 1066, 1025, 1007 (C-O-C), 1601, 1510, 800, 756, 733 (C=C). HR-MS, *m*/*z* (I_rel__._, %): 600 (3), 398 (15), 397 (60), 396 (95), 283 (34), 282 (100), 159 (98), 139.1 (71), 69 (15). Calcd. C_38_H_48_O_6_, *m*/*z* [M]^+^ 600.3436. Found 600.3448.

**Compound 7**. (4*S*,5′*R*,6a*R*,6b*S*,8a*S*,8b*R*,9*S*,10*R*,11a*S*,12a*S*,12b*S*)-5′,6a,8a,9-Tetramet- hyl-1,3,3′,4,4′,5,5′,6,6a,6b,6′,7,8,8a,8b,9,11a,12,12a,12b-icosahydrospiro(naphtho[2′,1′:4,5]indeno[2,1-*b*]furan-10,2′-pyran)-4-yl 4-(4-fluorophenyl)-2-oxobut-3-ynoate [(22*R*,25*R*)-spirost-5-en-3β-yl 4-(4-fluorophenyl)-2-oxobut-3-ynoate] (**7**). Yield 46%. Yellowish crystal. M.p. 62 °C (decomp.). [α]_D_^23^—105.3 (c 0.2, CHCl_3_). ^1^H NMR (400 MHz, CDCl_3_, δ, ppm): 0.74 (3H, d, *J* = 6.8 Hz, H-27), 0.77 (3H, s, H-19), 0.98 (3H, d, *J* = 7.0 Hz, H-21), 1.02 (1H, m, H-9), 1.07 (3H, s, H-18), 1.13–1.22 (3H, m, H-1,12,14), 1.30 (1H, m, H-15), 1.43–1.54 (3H, m, H-11,11,24), 1.58–1.69 (6H, m, H-7,8,23,23,24,25), 1.71–1.81 (3H, m, H-2,12,17), 1.88–1.92 (2H, m, H-1,20), 1.95–2.05 (3H, m, H-2,7,15), 2.48 (2H, m, H-4), 3.38 (1H, t, *J* = 10.2 Hz, H-26), 3.47 (1H, dm, *J* = 10.2 Hz, H-26), 4.41 (1H, dd, *J* = 14.8, 7.4 Hz, H-16), 4.81 (1H, m, H-3), 5.43 (1H, s, H-6), 7.13 (2H, t, *J* = 8.6 Hz, H-3″,5″), 7.68 (2H, dd, *J* = 8.6, 5.4 Hz, H-2″,6″). ^13^C NMR (126 MHz, CDCl_3_, δ, ppm): 14.5 (C-21), 16.2 (C-18), 17.1 (C-27), 19.3 (C-19), 20.8 (C-11), 27.4 (C-2), 28.8 (C-24), 30.3 (C-25), 31.4 (C-8), 31.4 (C-23), 31.8 (C-15), 32.1 (C-7), 36.8 (C-10), 36.9 (C-1), 37.7 (C-4), 39.7 (C-12), 40.3 (C-13), 41.6 (C-20), 49.9 (C-9), 56.5 (C-14), 62.2 (C-17), 66.9 (C-26), 77.5 (C-3), 80.8 (C-16), 87.2 (C-3′), 96.7 (C-4′), 109.3 (C-22), 115.3, 115.4 (C-1″), 116.3, 116.5 (C-3″,5″), 123.2 (C-6), 136.1, 136.2 (C-2″,6″), 138.9 (C-5), 158.7 (C-1′), 164.6 (d, C-4″, J_C-F_ = 255.7 Hz), 169.8 (C-2′). IR (KBr, ν, cm^−1^): 2191 (C≡C), 1740, 1680 (C=O), 1342, 1248, 1176, 1159 (C-F), 1082, 1066, 1024, 1007 (C-O-C), 1598, 1500, 815, 798, 760, 688 (C=C). HR-MS, *m*/*z* (I_rel__._, %): 588 (2), 397 (40), 396 (78), 283 (31), 282 (100), 281 (16), 253 (18), 147 (18), 139 (54). Calcd. C_37_H_45_FO_5_, *m*/*z* [M]^+^ 588.3246. Found 588.3242.

**Compound 8**. (4*S*,5′*R*,6a*R*,6b*S*,8a*S*,8b*R*,9*S*,10*R*,11a*S*,12a*S*,12b*S*)-5′,6a,8a,9-Tetramet- hyl-1,3,3′,4,4′,5,5′,6,6a,6b,6′,7,8,8a,8b,9,11a,12,12a,12b-icosahydrospiro(naphtho[2′,1′:4,5]indeno[2,1-*b*]furan-10,2′-pyran)-4-yl 2-oxo-4-(*m*-tolyl)but-3-ynoate [(22*R*,25*R*)-spirost- 5-en-3β-yl 2-oxo-4-(*m*-tolyl) but-3-ynoate] (**8**). Yield 30%. Yellowish solid. M.p. 148 °C (decomp.). [α]_D_^23^—50.9 (0.3, CHCl_3_). ^1^H NMR (300 MHz, CDCl_3_, δ, ppm): 0.76 (3H, d, *J* = 6.8 Hz, H-27), 0.77 (3H, s, H-19), 0.98 (3H, d, *J* = 7.0 Hz, H-21), 1.01 (1H, m, H-9), 1.08 (3H, s, H-18), 1.12–1.34 (4H, m, H-1,12,14,15), 1.43–1.69 (9H, m, H-7,8,11,11,23,23,24,24,25), 1.71–1.81 (3H, m, H-2,12,17), 1.84–1.93 (2H, m, H-1,20), 1.94–2.05 (3H, m, H-2,7,15), 2.38 (3H, s, C^7″^H_3_), 2.49 (2H, m, H^4,4^), 3.38 (1H, t, *J* = 10.8 Hz, H-26), 3.47 (1H, dd, *J* = 10.6, 2.6 Hz, H-26), 4.41 (1H, dd, *J* = 14.9, 7.5 Hz, H-16), 4.82 (1H, m, H-3), 5.43 (1H, d, *J* = 4.9 Hz, H-6), 7.22 (1H, d, *J* = 7.8 Hz, H-4″), 7.28 (1H, c, H-2″), 7. 30 (1H, d, *J* = 7.8 Hz, H-6″), 7.49 (1H, t, *J* = 7.8 Hz, H-6″). ^13^C NMR (126 MHz, CDCl_3_, δ, ppm): 14.5 (C-21), 16.2 (C-18), 17.1 (C-27), 19.3 (C-19), 20.8 (C-11), 21.1 (C-7″), 27.4 (C-2), 28.8 (C-24), 30.3 (C-25), 31.4 (C-8), 31.4 (C-23), 31.8 (C-15), 32.1 (C-7), 36.7 (C-10), 36.9 (C-1), 37.7 (C-4), 39.7 (C-12), 40.3 (C-13), 41.6 (C-20), 49.9 (C-9), 56.5 (C-14), 62.2 (C-17), 66.8 (C-26), 77.4 (C-3), 80.8 (C-16), 87.0 (C-3′), 98.3 (C-4′), 109.2 (C-22), 119.0 (C-1″), 123.1 (C-6), 128.6 (C-5″), 130.9 (C-4″), 132.7 (C-2″), 134.2 (C-6″), 138.6 (C-3″), 139.1 (C-5), 158.8 (C-1′), 169.9 (C-2′). IR (KBr, ν, cm^−1^): 2195 (C≡C), 1740, 1678 (C=O), 1080, 1066, 1025, 1005 (C-O-C), 1596, 1500, 815, 785, 756, 733, 687 (C=C). HR-MS, *m*/*z* (I_rel__._, %): 584 (1), 397 (32), 396 (65), 394 (27), 282 (49), 143 (100), 139 (96), 105 (16), 91 (17), 55 (19). Calcd. C_38_H_48_O_5_, *m*/*z* [M]^+^ 584.3496. Found 584.3495.

##### Four Component Reactions of (22*R*,25*R*)-Spirost-5-en-3β-yl 2-Chloro-2-Oxoacetate (**2**), Terminal Aryl Acetylenes (**3a**–**e**) and Phenylhydrazine Hydrochloride (**9**)

(a) A solution of diosgenin (800 mg, 1.93 mmol) in CHCl_3_ (10 mL) was added dropwise to a cold stirred solution of oxalyl chloride (980 mg (0.66 mL), 7.72 mmol) in CHCl_3_ (5 mL) in argon atmosphere at 0 °C for 1 h. The reaction mixture was stirred at room temperature for 3 h. The solvent was removed under reduced pressure, the residue was treated with 3ml of CHCl_3_ and additionally evaporated. This procedure was repeated three times for removing the trace of oxalyl chloride. The residue of **2** (975 mg) was dissolved in C_6_H_6_ (13 mL) and subsequently treated with aryl acetylene **3a**–**e** (1.93 mmol), CuI (36 mg, 0.19 mmol) and Et_3_N (195 mg (0.27 mL), 1.93 mmol), under stirring in an argon flow. The reaction mixture was stirred at 40 °C for 12h (TLC), then phenylhydrazine hydrochloride **9** (185 mg, 1.29 mmol) and Et_3_N (130 mg, 0.18 mL, 1.29 mmol) were added and temperature was increased to 60 °C. After 12 h stirring in argon atmosphere (TLC) the solvent was removed and the residue was purified by column chromatography (petroleum ether–ether, 100:15) to afford the corresponding compounds **10**–**14**. (b) A solution of diosgenin (800 mg, 1.93 mmol) in CHCl_3_ (10 mL) was added dropwise to a cold stirred solution of oxalyl chloride (980 mg (0.66 mL), 7.72 mmol) in CHCl_3_ (5 mL) in argon atmosphere at 0 °C for 1 h. The reaction mixture was additionally stirred at room temperature for 3 h. The solvent was removed under reduced pressure, the residue was treated with 3ml of CHCl_3_ and additionally evaporated. This procedure was repeated three times for removing the trace of oxalyl chloride and the residue was dissolved in C_6_H_6_ (13 mL) in an argon flow. The corresponding terminal aryl acetylene **3a**–**e** (1.93 mmol), CuI (36 mg, 0.19 mmol), and Et_3_N (195 mg (0.27 mL), 1.93 mmol) were added. The reaction mixture was stirred at 40 °C for 12h (TLC), then compound **9** (185 mg,1.29 mmol), ethanol (7mL), and Et_3_N (130 mg, 0.18 mL, 1.29 mmol), were added and the temperature was increased to 60 °C. After 12 h stirring under argon atmosphere (TLC) the solvent was removed and the residue was subjected to column chromatography (eluent petroleum ether–ether, gradient from 20:1 to 100:15) with sequentional isolation of the steroidal (*E*)-alkynylhydrazones **15**–**19** and pyrazoles **10**–**14**. (c) A solution of diosgenin **1** (800 mg, 1.93 mmol) in CHCl_3_ (10 mL) was added dropwise to a cold stirred solution of oxalyl chloride (980 mg (0.66 mL), 7.72 mmol) in CHCl_3_ (5 mL) in argon atmosphere at 0 °C for 1 h. The reaction mixture was stirred at room temperature for 3 h. The solvent was removed under reduced pressure, the residue was treated with 3ml of CHCl_3_ and additionally evaporated. This procedure was repeated three times for removing the trace of oxalyl chloride and the formed compound **2**, aryl acetylene **3a** (197 mg, 1.93 mmol), CuI (36 mg, 0.19 mmol) and Et_3_N (195 mg (0.27 mL), 1.93 mmol) was stirred in C_6_H_6_ (13 mL) in an argon flow at 40 °C for 12 h (TLC). The solvent was removed under reduced pressure, and the residue was treated with 2-methoxyethanol (15 mL). Then, phenylhydrazine hydrochloride **9** (223 mg, 1.54 mmol) and Et_3_N (156 mg (0.21 mL), 1.54 mmol) were added in an argon flow. The reaction mixture was stirred for 24 h at ambient temperature, then the solvent was evaporated. Column chromatography of the residue (eluent petroleum ether–ether, gradient from 100:3 to 100:15) afforded (*Z*)-alkynylhydrazone **20**, (*E*)-alkynylhydrazone **15**, pyrazole **21** and pyrazole **10**.

**Compound 10**. (4*S*,5′*R*,6a*R*,6b*S*,8a*S*,8b*R*,9*S*,10*R*,11a*S*,12a*S*,12b*S*)-5′,6a,8a,9-Tetra- methyl-1,3,3′,4,4′,5,5′,6,6a,6b,6′,7,8,8a,8b,9,11a,12,12a,12b-icosahydrospiro(naphtho[2′,1′: 4,5]indeno[2,1-*b*]furan-10,2′-pyran)-4-yl 1,5-diphenyl-1*H*-pyrazole-3-carboxylate [(22*R*,25*R*)-spirost-5-en-3β-yl 1,5-diphenyl-1*H*-pyrazole-3-carboxylate] (**10**). Yield: 46% (method a), 9% (method b), 12% (method c). White solid. M.p. 207–210 °C (petroleum and diethyl ethers mixture). [α]_D_^23^ –67.3 (c 0.4, CHCl_3_). ^1^H NMR (600 MHz, CDCl_3_, δ, ppm): 0.78 (3H, s, H-19), 0.75 (3H, d, *J* = 6.8 Hz, H-27), 0.98 (3H, d, *J* = 7.0 Hz, H-21), 1.02 (1H, m, H-9), 1.08 (3H, s, H-18), 1.14 (1H, m, H-14), 1.22 (2H, m, H-1,12), 1.30 (1H, m, H-15), 1.42–1.52 (2H, m, H-11,24), 1.54–1.71 (7H, m, H-7,8,11,23,23,24,25), 1.76 (1H, dm, *J* = 12.5 Hz, H-12), 1.79–1.85 (2H, m, H-2,17), 1.89 (1H, m, H-20), 1.92 (1H, dt, *J* = 13.8, 3.7 Hz, H-1), 1.99–2.05 (3H, m, H-2,7,15), 2.53 (2H, m, H-4,4), 3.39 (1H, t, J = 10.7 Hz, H-26), 3.48 (1H, dm, *J* = 10.7 Hz, H-26), 4.42 (1H, dd, *J* = 15.0, 7.3 Hz, H-16), 4.95 (1H, m, H-3), 5.43 (1H, d, *J* = 4.8 Hz, H-6), 7.04 (1H, s, H-4′), 7.21–7.35 (10H, m, H-2″, 3″, 4″,5″, 6″,2‴,3‴,4‴,5‴,6‴). ^13^C NMR (151 MHz, CDCl_3_, δ, ppm): 14.5 (C-21), 16.3 (C-18), 17.1 (C-27), 19.4 (C-19), 20.8 (C-11), 27.1 (C-2), 28.8 (C-24), 30.3 (C-25), 31.3 (C-23), 31.4 (C-8), 31.8 (C-15), 32.1 (C-7), 36.8 (C-10), 37.0 (C-1), 38.1 (C-4), 39.7 (C-12), 40.3 (C-13), 41.6 (C-20), 49.9 (C-9), 56.4 (C-14), 62.1 (C-17), 66.8 (C-26), 74.7 (C-3), 80.8 (C-16), 109.2 (C-22), 109.9 (C-4′), 122.5 (C-6), 125.8 (2C), 128.3, 128.5, 128.6, 128.7, 128.9, 129.0, 129.2 (2C) (C-2″,6″,3″,5″,4″,2‴,6‴,3‴, 5‴,4‴), 129.6 (C-1″), 139.6 (C-1‴), 139.8 (C-5), 144.5, 144.6 (C-3′,5′), 161.9 (C=O). IR (KBr, ν, cm^−1^): 1718 (C=O), 1535, 1569 (C=N pyrazole), 1078, 1065, 1049, 1026, 1009 (C-O-C), 1599, 1502, 829, 796, 762, 698 (C=C). HR-MS, *m*/*z* (I_rel._, %): 660 (1), 396 (100), 265 (92), 282 (66), 139 (63), 247 (33), 397 (32), 266 (18), 283 (17), 324 (14). Calcd. C_43_H_52_N_2_O_4_, *m*/*z* [M]^+^. 660.3922. Found 660.3909.

**Compound 11**. (4*S*,5′*R*,6a*R*,6b*S*,8a*S*,8b*R*,9*S*,10*R*,11a*S*,12a*S*,12b*S*)-5′,6a,8a,9-Tetra- methyl-1,3,3′,4,4′,5,5′,6,6a,6b,6′,7,8,8a,8b,9,11a,12,12a,12b-icosahydrospiro(naphtho[2′,1′: 4,5]indeno[2,1-*b*]furan-10,2′-pyran)-4-yl 5-(4-ethylphenyl)-1-phenyl-1*H*-pyrazole-3-car- boxylate [(22*R*,25*R*)-spirost-5-en-3β-yl 5-(4-ethylphenyl)-1-phenyl-1*H*-pyrazole-3-carbo- xylate] (**11**). Yield: 53% (a), 13% (b). White needles. M.p. 177–180 °C. [α]_D_^25^—61.9 (c 0.2, CHCl_3_). ^1^H NMR (400 MHz, CDCl_3_, δ, ppm): 0.78 (3H, s, H-19), 0.76 (3H, d, *J* = 6.8 Hz, H-27), 0.98 (3H, d, *J* = 7.0 Hz, H-21), 1.02 (1H, m, H-9), 1.08 (3H, s, H-18), 1.24 (3H, t, *J* = 7.2 Hz, H-8″), 1.20–1.34 (4H, m, H-1,12,14,15), 1.44–1.68 (9H, m, H-7,8,11,11,23,23,24,24,25), 1.72–1.82 (3H, m, H-2,12,17), 1.87–1.94 (2H, m, H-1,20), 1.98–2.05 (3H, m, H-2,7,15), 2.52 (2H, m, H-4), 2.63 (2H, q, *J* = 7.2 Hz, H-7″), 3.39 (1H, t, *J* = 10.9 Hz, H-26), 3.48 (1H, dm, *J* = 10.9 Hz, H-26), 4.42 (1H, dd, *J* = 14.5, 7.4 Hz, H-16), 4.95 (1H, m, H-3), 5.43 (1H, d, *J* = 3.8 Hz, H-6), 7.01 (1H, s, H-4′), 7.13–7.26 (4H, m, H-2″,6″,3″, 5″), 7.28–7.35 (5H, m, H-2‴,6‴,3‴,5‴,4‴). ^13^C NMR (126 MHz, CDCl_3_, δ, ppm): 14.5 (C-21), 15.2 (C-8″), 16.3 (C-18), 17.1 (C-27), 19.4 (C-19), 20.8 (C-11), 27.7 (C-2), 28.5 (C-7″), 28.8 (C-24), 30.3 (C-25), 31.3 (C-23), 31.4 (C-8), 31.8 (C-15), 32.0 (C-7), 36.8 (C-10), 36.9 (C-1), 38.1 (C-4), 39.7 (C-12), 40.2 (C-13), 41.6 (C-20), 49.9 (C-9), 56.4 (C-14), 62.0 (C-17), 66.8 (C-26), 74.6 (C-3), 80.8 (C-16), 109.2 (C-22), 109.7 (C-4′), 122.4 (C-6), 125.8 (2C), 128.0, 128.2 (2C), 128.6 (2C), 128.9 (2C) (C-2″,6″,3″,5″,2‴,6‴,3‴,5‴, 4‴), 126.6 (C-1″), 139.6 (C-1‴), 139.7 (C-5), 144.5, 144.6 (C-3′,5′), 144.9 (C-4″), 161.9 (C=O). IR (KBr, ν, cm^−1^): 1714 (C=O), 1535, 1556 (C=N pyrazole), 1066, 1053, 1024, 1007 (C-O-C), 1599, 1502, 825, 796, 764, 710, 692 (C=C). HR-MS, *m*/*z* (I_rel._, %): 688 (1), 397 (18), 396 (52), 294 (20), 293 (100), 292 (25), 282 (39), 275 (14), 139 (23). Calcd. C_45_H_56_N_2_O_4_. *m*/*z* [M]^+^ 688.4149. Found 688.4154. X-ray structural analysis of compound **11**: C_45_H_56_N_2_O_4_, M 688.92, monoclinic, P2_1_, a 13.4005(8), b 7.8696(5), c 19.590(1) Å, β 108.795(3)º, V 1955.7(2) Å^3^, Z 2, D_calcd_ 1.170g·cm^−3^, μ(Mo-Kα) 0.074 mm^−1^, F(000) 744, (θ 1.10–25.06°, completeness 99.7%), colorless, (0.64 × 0.20 × 0.05) mm^3^, transmission 0. 0.7514–0.8620. The intensities of 6917 independent reflections were measured (R_int_ 0.0393), 465 parameters, 7 restraints, R_1_ 0.0557 (for 6319 observed I > 2σ(I)), wR_2_ = 0.1766 (all data), GOOF 1.067, largest diff. peak and hole 0.182 and -0.207 e.A^−3^. Crystallographic data for structure **11** have been deposited at the Cambridge Crystallographic Data Centre as supplementary publication no. CCDC 2077784.

**Compound 12**. (4*S*,5′*R*,6a*R*,6b*S*,8a*S*,8b*R*,9*S*,10*R*,11a*S*,12a*S*,12b*S*)-5′,6a,8a,9-Tetra- methyl-1,3,3′,4,4′,5,5′,6,6a,6b,6′,7,8,8a,8b,9,11a,12,12a,12b-icosahydrospiro(naphtho[2′,1′: 4,5]indeno[2,1-*b*]furan-10,2′-pyran)-4-yl 5-(4-methoxyphenyl)-1-phenyl-1*H*-pyra- zole-3-carboxylate [(22*R*,25*R*)-spirost-5-en-3*β*-yl 5-(4-methoxyphenyl)-1-phenyl- 1*H*-pyrazole-3-carboxylate] (**12**). Yield: 54% (*a*), 17% (*b*). Yellowish needles. M.p. 223–224 °C (decomp.). [α]_D_^22^ -61.9 (*c* 0.2, CHCl_3_). ^1^H NMR (400 MHz, CDCl_3_, δ, ppm): 0.77 (3H, s, H-19), 0.76 (3H, d, *J* = 6.8 Hz, H-27), 0.98 (3H, d, *J* = 6.9 Hz, H-21), 1.01 (1H, m, H-9), 1.08 (3H, c, H-18), 1.12–1.34 (4H, m, H-1,12,14,15), 1.43–1.68 (9H, m, H-7,8,11,11,23, 23,24,24,25), 1.71–1.82 (3H, m, H-2,12,17), 1.85–1.93 (2H, m, H-1,20), 1.97–2.05 (3H, m, H-2,7,15), 2.52 (2H, m, H-4), 3.39 (1H, t, *J* = 10.9 Hz, H-26), 3.48 (1H, dm, *J* = 10.9, H-26), 3.77 (3H, s, OCH_3_), 4.42 (1H, dd, *J* = 15.0, 7.7 Hz, H-16), 4.94 (1H, m, H-3), 5.42 (1H, d, *J* = 5.0, H-6), 6.78 (2H, d, *J* = 8.7 Hz, H-3″,5″), 6.94 (1H, s, H-4′), 7.12 (2H, d, *J* = 8.7 Hz, H-2″,6″), 7.26–7.35 м (5H, H-2‴,3‴,4‴,5‴,6‴). ^13^C NMR (126 MHz, CDCl_3_, δ, ppm): 14.5 (C-21), 16.3 (C-18), 17.1 (C-27), 19.4 (C-19), 20.8 (C-11), 27.7 (C-2), 28.8 (C-24), 30.3 (C-25), 31.3 (C-23), 31.4 (C-8), 31.8 (C-15), 32.0 (C-7), 36.8 (C-10), 36.9 (C-1), 38.0 (C-4), 39.7 (C-12), 40.2 (C-13), 41.6 (C-20), 49.9 (C-9), 55.2 (OCH_3_), 56.4 (C-14), 62.0 (C-17), 66.8 (C-26), 74.6 (C-3), 80.8 (C-16), 109.2 (C-22), 109.4 (C-4′), 113.9 (C-3″,5″), 121.92 (C-1″), 122.4 (C-6), 125.8 (2C), 128.2, 128.9 (2C) (C-2‴,6‴,3‴,5‴,4‴), 130.0 (C-2″,6″), 139.6 (C-1‴), 139.7 (C-5), 144.4, 144.5 (C-3′,5′), 159.8 (C-4″), 161.9 (C=O). IR (KBr, ν, cm^−1^): 1738 (C=O), 1533, 1552, 1578 (C=N pyrazole), 1066, 1051, 1028, 1007 (C-O-C), 1612, 1500, 812, 796, 777, 762, 692 (C=C). HR-MS, *m/z* (I_rel._, %): 690 (1), 397 (12), 396 (41), 296 (14), 295 (100), 294 (53), 282 (32), 277 (28), 139 (39). Calcd. C_44_H_54_N_2_O_5_, *m/z* [M]^+^ 690.4027. Found 690.4094. X-ray structural analysis of compound **12**: C_44_H_54_N_2_O_5_, *M* 690.89, monoclinic, *P2_1_*, *a* 13.3267(6), *b* 7.8178(3), *c* 19.2083(9) Å, *β* 108.337(2)º, *V* 1899.6(1) Å^3^, *Z* 2, *D*_calcd_ 1.208 g·cm^−3^, *μ*(Mo-*K*α) 0.078 mm^−1^, F(000) 744, (θ 2.23–25.02°, completeness 99.9%), colorless, (0.64 × 0.20 × 0.05) mm^3^, transmission 0.8295–0.8620. The intensities of 6691 independent reflections were measured (*R*_int_ 0.0322), 465 parameters, *R*_1_ 0.0355 (for 5873 observed *I >* 2*σ*(*I*)), *wR*_2_ = 0.1119 (all data), GOOF 1.091, largest diff. peak and hole 0.167 and -0.197 e.A^−3^. Crystallographic data for structure **12** have been deposited at the Cambridge Crystallographic Data Centre as supplementary publication no. CCDC 2077782.

**Compound 13**. (4*S*,5′*R*,6a*R*,6b*S*,8a*S*,8b*R*,9*S*,10*R*,11a*S*,12a*S*,12b*S*)-5′,6a,8a,9-Tetra- methyl-1,3,3′,4,4′,5,5′,6,6a,6b,6′,7,8,8a,8b,9,11a,12,12a,12b-icosahydrospiro(naphtho[2′,1′: 4,5]indeno[2,1-*b*]furan-10,2′-pyran)-4-yl 5-(4-fluorophenyl)-1-phenyl-1*H*-pyrazole-3- carboxylate [(22*R*,25*R*)-spirost-5-en-3β-yl 5-(4-fluorophenyl)-1-phenyl-1*H*-pyrazole- 3-carboxylate] (**13**). Yield: 48% (a), 69% (b). Yellowish crystals. M.p. 223 °C (decomp.). [α]_D_^22^—52.2 (0.23, CHCl_3_). ^1^H NMR (400 MHz, CDCl_3_, δ, ppm): 0.77 (3H, s, H-19), 0.75 (3H, d, *J* = 7.0 Hz, H-27),0.98 (3H, d, *J* = 6.9 Hz, H-21), 1.01 (1H, m, H-9), 1.08 (3H, s, H-18), 1.11–1.34 (4H, m, H-1,12,14,15), 1.44–1.68 (9H, m, H-7,8,11,11,23,23,24,24,25), 1.71–1.81 (3H, m, H-2,12,17), 1.85–1.94 (2H, m, H-1,20), 1.97–2.05 (3H, m, H-2,7,15), 2.52 (2H, m H-4), 3.39 (1H, t, *J* = 10.9 Hz, H-26), 3.48 (1H, dm, *J* = 10.9 Hz, H-26), 4.42 (1H, dd, *J* = 14.9, 7.4 Hz, H-16), 4.95 (1H, m, H-3), 5.43 (1H, d, *J* = 4.0 Hz, H-6), 6.98–7.02 (3H, m, H-4′,3″,5″), 7.18 (2H, dd, *J* = 8.5, 5.4 Hz, H-2″,6″), 7.29–7.36 (5H, m, H-2‴,6‴,3‴,5‴,4‴). ^13^C NMR (101 MHz, CDCl_3_,) δ, ppm): 14.5 (C-21), 16.3 (C-18), 17.1 (C-27), 19.4 (C-19), 20.8 (C-11), 27.7 (C-2), 28.76 (C-24), 30.26 (C-25), 31.35 (C-23), 31.39 (C-8), 31.81 (C-15), 32.04 (C-7), 36.8 (C-10), 36.98 (C-1), 38.04 (C-4), 39.69 (C-12), 40.23 (C-13), 41.57 (C-20), 49.89 (C-9), 56.4 (C-14), 62.03 (C-17), 66.80 (C-26), 74.69 (C-3), 80.76 (C-16), 109.24 (C-22), 109.91 (C-4′), 115.6, 115.8 (C-3″,5″), 122.5 (C-6), 125.7 (C-1″), 125.8 (C-2‴,6‴), 128.4, (C-3‴,5‴), 129.1 (C-4‴), 130.5, 130.6 (C-2″,6″), 139.3 (C-1‴), 139.7 (C-5), 144.1, 144.6 (C-3′,5′), 162.7 (C-4″, J_C-F_ = 257.8 Hz), 161.8 (C=O). IR (KBr, ν, cm^−1^): 1718 (C=O), 1531, 1554 (C=N pyrazole), 1342, 1227, 1194, 1159 (C-F), 1068, 1051, 1026, 1007 (C-O-C), 1610, 1500, 816, 796, 777, 758, 692 (C=C). HR-MS, *m*/*z* (I_rel_, %): 678 (1), 397 (26), 396 (84), 283 (25), 282 (100), 265 (33), 139 (95), 69 (26). Calcd.C_43_H_51_FN_2_O_4_, *m*/*z* [M]^+^ 678.3827. Found 678.3828.

**Compound 14**. (4*S*,5′*R*,6a*R*,6b*S*,8a*S*,8b*R*,9*S*,10*R*,11a*S*,12a*S*,12b*S*)-5′,6a,8a,9-Tetra- methyl-1,3,3′,4,4′,5,5′,6,6a,6b,6′,7,8,8a,8b,9,11a,12,12a,12b-icosahydrospiro(naphtho[2′,1′: 4,5]indeno[2,1-*b*]furan-10,2′-pyran)-4-yl 1-phenyl-5-(*m*-tolyl)-1*H*-pyrazole-3-carboxylate [(22*R*,25*R*)-spirost-5-en-3β-yl 1-phenyl-5-(*m*-tolyl)-1*H*-pyrazole-3-carboxylate] (**14**). Yield: 60% (a), 39% (b). Red needles. M.p. 101 °C (decomp.). [α]_D_^23^—56.6 (c 0.2, CHCl_3_). ^1^H NMR (300 MHz, CDCl_3_, δ, ppm): 0.78 (3H, s, H-19), 0.75 (3H, d, *J* = 7.0 Hz, H-27), 0.98 (3H, d, *J* = 6.9 Hz, H-21), 1.02 (1H, m, H-9), 1.08 (3H, s, H-18), 1.12–1.34 (4H, m, H-1,12,14,15), 1.44–1.69 (9H, m, H-7,8,11,11,23,23,24,24,25), 1.72–1.83 (3H, m, H-2,12,17), 1.85–1.94 (2H, m, H-1,20), 1.98–2.05 (3H, m, H-2,7,15), 2.29 (3H, s, H-7″) 2.53 (2H, m, H-4), 3.39 (1H, t, J = 10.9 Hz, H-26), 3.48 (1H, dm, *J* = 10.9 Hz, H-26), 4.42 (1H, dd, *J* = 14.9, 7.1 Hz, H-16), 4.95 (1H, m, H-3), 5.43 (1H, *J* = 4.9 Hz, H-6), 6.87 (1H, *J* = 7.3 Hz, H-6″), 7.02 (1H, s, H-4′), 7.08 (1H, br.s, H-2″), 7.09–7.21 (2H, m, H-4″,5″), 7.29–7.39 (5H, m, H-2‴,6‴,3‴,5‴,4‴). ^13^C NMR (75 MHz, CDCl_3_, δ, ppm): 14.5 (C-21), 16.3 (C-18), 17.1 (C-27), 19.4 (C-19), 20.8 (C-11), 21.3 (C-7″), 27.7 (C-2), 28.8 (C-24), 30.3 (C-25), 31.4 (C-23), 31.4 (C-8), 31.8 (C-15), 32.1 (C-7), 36.8 (C-10), 37.0 (C-1), 38.1 (C-4), 39.7 (C-12), 40.2 (C-13), 41.6 (C-20), 49.9 (C-9), 56.4 (C-14), 62.1 (C-17), 66.8 (C-26), 74.6 (C-3), 80.8 (C-16), 109.3 (C-22), 109.9 (C-4′), 115.68 (C-2″), 122.45 (C-6), 125.74, 125.83, 128.23, 128.3, 128.87, 129.38 (C-4″,5″,6″,2‴,6‴,3‴,5‴, 4‴), 129.49 (C-3″), 138.29 (C-1″), 139.6 (C-1‴), 139.8 (C-5), 144.5, 144.7 (C-3′,5′), 161.9 (C=O). IR (KBr, ν, cm^−1^): 1716 (C=O), 1528, 1562 (C=N pyrazole), 1070, 1053, 1026, 1007 (C-O-C), 1599, 1500, 825, 777, 762, 700 (C=C). HR-MS, *m*/*z* (I_rel._, %): 674 (0.4), 396 (54), 397 (28), 283 (20), 282 (74), 143 (77), 139 (100), 105 (23), 91 (26), 69 (23). Calcd. C_44_H_54_N_2_O_4_, *m*/*z* [M]^+^ 674.4078. Found 674.4073.

**Compound 15**. (4*S*,5′*R*,6a*R*,6b*S*,8a*S*,8b*R*,9*S*,10*R*,11a*S*,12a*S*,12b*S*)-5′,6a,8a,9-Tetra- methyl-1,3,3′,4,4′,5,5′,6,6a,6b,6′,7,8,8a,8b,9,11a,12,12a,12b-icosahydrospiro(naphtho[2′,1′: 4,5]indeno[2,1-*b*]furan-10,2′-pyran)-4-yl (*E*)-4-phenyl-2-(2-phenylhydrazineylidene)- but-3-ynoate [(22*R*,25*R*)-spirost-5-en-3β-yl (*E*)-4-phenyl-2-(2-phenylhydrazineylidene)- but-3-ynoate] (**15**). Yield 35% (b), 47% (c). Yellow needles. M.p. 165 °C (decomp.). [α]_D_^23^ -59.0 (c 0.27, CHCl_3_). ^1^H NMR (300 MHz, CDCl_3_, δ, ppm): 0.78 (3H, s, H-19), 0.77 (3H, d, *J* = 6.8 Hz, H-27), 0.99 (3H, d, *J* = 6.7 Hz, H-21), 1.04 (1H, m, H-9), 1.10 (3H, s, H-18), 1.14–1.22 (2H, m, H-1,14), 1.24–1.36 (2H, m, H-12,15), 1.43–1.97 (14H, m, H-1,2,7,8,11,11,12, 17,20,23,23,24,24,25), 1.99–2.08 (3H, m, H-2,7,15), 2.52 (2H, m, H-4,4), 3.39 (1H, t, *J* = 10.8 Hz, H-26), 3.49 (1H, dd, *J* = 10.8, 3.1 Hz, H-26), 4.42 (1H, dd, *J* = 14.8, 7.6 Hz, H-16), 4.83 (1H, m, H-3), 5.43 (1H, d, *J* = 4.4 Hz, H-6). 7.05 (1H, t, *J* = 7.0 Hz, H-4″), 7.28–7.45, 7.59–7.62 (9H, m, H-2″,6″,3″,5″,2‴,6‴,3‴,5‴,4‴), 9.09 (1H, s, NH). ^13^C NMR (75 MHz, CDCl_3_, δ, ppm): 14.5 C-21), 16.3 (C-18), 17.1 (C-27), 19.4 (C-19), 20.8 (C-11), 27.1 (C-2), 28.8 (C-24), 30.3 (C-25), 31.3 (C-23), 31.4 (C-8), 31.8 (C-15), 32.1 (C-7), 36.8 (C-10), 36.9 (C-1), 38.1 (C-4), 39.7 (C-12), 40.2 (C-13), 41.5 (C-20), 49.9 (C-9), 56.4 (C-14), 62.1 (C-17), 66.8 (C-26), 75.3 (C-3), 77.7 (C-3′), 80.8 (C-16), 104.8 (C-4′), 109.2 (C-22), 114.8 (C-4″), 117.8 (C-2′), 121.4 (C-1″), 122.5 (C-6), 123.4 (C-4‴), 128.6, 129.4, 129.6, 131.9 (2C-2″,6″,3″,5″,2‴,6‴,3‴,5‴), 139.74 (C-5), 141.8 (C-1‴), 162.3 (C-1′). IR (KBr, ν, cm^−1^): 3282 (NH), 2181 (C≡C), 1711 (C=O), 1531 (C=N), 1070, 1049, 1028, 1009 (C-O-C), 1603, 1502, 812, 802, 754, 733, 716, 690 (C=C). HR-MS, *m*/*z* (I_rel_, %): 660 (10), 139 (100), 282 (77), 396 (77), 193 (50), 265 (48), 397 (37), 264 (33), 115 (32), 324 (30). Found, *m*/*z*: 660.3913 [M]^+^. C_43_H_52_N_2_O_4_. Calculated, *m*/*z*: 660.3922. X-ray structural analysis of compound **15**: C_43_H_52_N_2_O_4_, M 660.87, orthorhombic, P2_1_2_1_2_1_, a 6.859(1), b 11.538(2), c 47.200(7) Å, V 3735.1(9) Å^3^, Z 4, D_calcd_ 1.175 g·cm^−3^, μ(Mo-Kα) 0.075 mm^−1^, F(000) 1424, (θ 0.86–25.25°, completeness 99.5%), colorless, (0.75 × 0.14 × 0.05) mm^3^, transmission 0.7744–0.8620. The intensities of 6536 independent reflections were measured (R_int_ 0.0647), 446 parameters, R_1_ 0.0484 (for 5539 observed I > 2σ(I)), wR_2_ = 0.1532 (all data), GOOF 0.947, largest diff. peak and hole 0.226 and -0.236 e.A^−3^. Crystallographic data for structure **15** have been deposited at the Cambridge Crystallographic Data Centre as supplementary publication no. CCDC 2077783.

**Compound 16**. (4*S*,5′*R*,6a*R*,6b*S*,8a*S*,8b*R*,9*S*,10*R*,11a*S*,12a*S*,12b*S*)-5′,6a,8a,9-Tetra- methyl-1,3,3′,4,4′,5,5′,6,6a,6b,6′,7,8,8a,8b,9,11a,12,12a,12b-icosahydrospiro(naphtho[2′,1′: 4,5]indeno[2,1-*b*]furan-10,2′-pyran)-4-yl (*E*)-4-(4-ethylphenyl)-2-(2-phenylhydrazineyli- dene)-but-3-ynoate [(22*R*,25*R*)-spirost-5-en-3β-yl [(*E*)-4-(4-ethylphenyl)-2-(2-phenylhyd- razineylidene)-but-3-ynoate] (**16**). Yield 37% (b). Yellow needles. M.p. 98 °C (decomp.). [α]_D_^20^ -49.6 (c 0.2, CHCl_3_). ^1^H NMR (600 MHz, CDCl_3_, δ, ppm): 0.78 (3H, s, H-19), 0.76 (3H, d, *J* = 6.8 Hz, H-27), 0.99 (3H, d, *J* = 7.0, H-21), 1.02 (1H, m, H-9), 1.10 (3H, s, H-18), 1.14 (1H, m, H-14), 1.22 (2H, m, H-1,12), 1.27 (3H, t, *J* = 7.7 Hz, H-C^8″^H_3_), 1.30 (1H, m, H-15), 1.48 (2H, m, H-11,24), 1.55–1.77 (7H, m, H-7,8,11,23,23,24,25), 1.75–1.77 (3H, m, H-2,12,17), 1.86–1.93 (2H, m, H-1,20), 1.99–2.05 (3H, m, H-2,7,15), 2.51 (2H, m, H-4), 2.70 (2H, q, *J* = 7.7 Hz, H-7″), 3.39 (1H, t, *J* = 11.0 Hz, H-26), 3.49 (1H, dm, *J* = 11.0 Hz, H-26), 4.43 (1H, dd, *J* = 15.4, 7.7 Hz, H-16), 4.83 (1H, m, H-3), 5.42 (1H, d, *J* = 5.1 Hz, H-6), 7.05 (1H, t, *J* = 7.3 Hz, H-4‴), 7.25 (2H, d, *J* = 8.1 Hz, H-3″,5″), 7.29 (2H, d, *J* = 7.7 Hz, H-2‴,6‴), 7.34 (2H, t, *J* = 7.3 Hz, H-3‴,5‴), 7.51 (2H, d, *J* =8.1 Hz, H-2″,6″), 9.08 (1H, s, NH). ^13^C NMR (151 MHz, CDCl_3_, δ, ppm): 14.5 (C-21), 15.3 (C-8″), 16.3 (C-18), 17.1 (C-27), 19.4 (C-19), 20.8 (C-11), 27.7 (C-2), 28.8 (C-24), 28.9 (C-7″), 30.3 (C-25), 31.3 (C-23), 31.5 (C-8), 31.8 (C-15), 32.1 (C-7), 36.8 (C-10), 37.0 (C-1), 38.1 (C-4), 39.7 (C-12), 40.3 (C-13), 41.6 (C-20), 49.9 (C-9), 56.4 (C-14), 62.1 (C-17), 66.8 (C-26), 75.3 (C-3), 77.3 (C-3′), 80.8 (C-16), 105.2 (C-4′), 109.3 (C-22), 114.8 (C-2‴,6‴), 118.0 (C-2′), 118.5 (C-1″), 122.4 (C-6), 123.2 (C-4‴), 128.2 (C-3″,5″), 129.4 (C-3‴,5‴), 131.9 (C-2″,6″), 139.8 (C-5), 141.9 (C-1‴), 146.4 (C-4″), 162.4 (C-1′). IR (KBr, ν, cm^−1^): 3279 (NH), 2187 (C≡C), 1711 (C=O), 1559 (C=N), 1068, 1051, 1028, 1007 (C-O-C), 1602, 1510, 831, 800, 752, 710, 690 (C=C). HR-MS, *m*/*z* (I_rel._, %): 688 (1), 396 (50), 294 (20), 293 (100), 292 (24), 282 (46), 275 (56), 139 (77), 69 (33), 55 (20). Calcd. C_45_H_56_N_2_O_4_, *m*/*z* [M]^+^ 688.4235. Found 688.4233.

**Compound 17**. (4*S*,5′*R*,6a*R*,6b*S*,8a*S*,8b*R*,9*S*,10*R*,11a*S*,12a*S*,12b*S*)-5′,6a,8a,9-Tetra- methyl-1,3,3′,4,4′,5,5′,6,6a,6b,6′,7,8, 8a,8b,9,11a,12,12a,12b-icosahydrospiro[naphtho(2′,1′: 4,5]indeno[2,1-*b*]furan-10,2′-pyran)-4-yl (*E*)-4-(4-methoxyphenyl)-2-(2-phenylhydrazine- ylidene)but-3-ynoate [(22*R*,25*R*)-spirost-5-en-3β-yl (E)-4-(4-methoxyhenyl)-2-(2-phenyl- hydrazineylidene)but-3-ynoate] (**17**). Yield 38% (b). Yellow solid. M.p. 111 °C (decomp.). [α]_D_^22^ -53.1 (c 0.2, CHCl_3_). ^1^H NMR (600 MHz, CDCl_3_, δ, ppm): 0.76 (3H, s, H-19), 0.77 (3H, d, *J* = 7.0 Hz, H-27), 0.98 (3H, d, *J* = 7.0 Hz, H-21), 1.02 (1H, m, H-9), 1.09 (3H, s, H-18), 1.14 (1H, m, H-14), 1.21 (2H, m, H-1,12), 1.30 (1H, m, H-15), 1.48 (2H, m, H-11,24), 1.54–1.71 (7H, m, H-7,8,11,23,23,24,25), 1.74–1.81 (3H, m, H-2,12,17), 1.86–1.93 (2H, m, H-1,20), 1.99–2.06 (3H, m, H-2,7,15), 2.52 (2H, m, H-4), 3.39 (1H, t, *J* = 10.8 Hz, H-26), 3.48 (1H, dm, *J* = 10.8 Hz, H-26), 3.86 (3H, s, H-OCH_3_), 4.42 (1H, dd, *J* =15.0, 7.3 Hz, H-16), 4.82 (1H, m, H-3), 5.42 (1H, d, *J* = 5.1 Hz, H-6), 6.93 (2H, d, *J* = 8.8 Hz, H-3″,5″), 7.04 (1H, t, *J* = 7.3 Hz, H-4‴), 7.28 (2H, d, *J* = 8.2 Hz, H-2‴,6‴), 7.34 (2H, dd, *J* = 8.2, 7.3 Hz, H-3‴,5‴), 7.53 (2H, d, *J* = 8.8 Hz, H-2″,6″), 9.06 (1H, s, NH). ^13^C NMR (151 MHz, CDCl3, δ, ppm): 14.5 (C-21), 16.3 (C-18), 17.1 (C-27), 19.4 (C-19), 20.8 (C-11), 27.7 (C-2), 28.8 (C-24), 30.3 (C-25), 31.3 (C-23), 31.5 (C-8), 31.8 (C-15), 32.1 (C-7), 36.8 (C-10), 37.0 (C-1), 38.1 (C-4), 39.7 (C-12), 40.2 (C-13), 41.6 (C-20), 49.9 (C-9), 55.4 (OCH_3_), 56.4 (C-14), 62.1 (C-17), 66.8 (C-26), 75.3 (C-3), 76.8 (C-3′), 80.8 (C-16), 105.2 (C-4′), 109.2 (C-22), 113.4 (C-1″), 114.3 (C-3″,5″), 114.7 (C-2‴,6‴), 118.2 (C-2′), 122.4 (C-6), 123.1 (C-4‴), 129.4 (C-3‴,5‴), 133.5 (C-2″,6″), 139.8 (C-5), 141.9 (C-1‴), 160.7 (C-4″), 162.4 (C-1′). IR (KBr, ν, cm^−1^): 3280 (NH), 2191 (C≡C), 1711 (C=O), 1527 (C=N), 1068, 1051, 1026, 1007 (C-O-C), 1603, 1500, 833, 797, 785, 750, 710, 690 (C=C). HR-MS, *m*/*z* (I_rel._, %): 690 (1), 397 (14), 396 (46), 295 (100), 294 (59), 277 (34), 282 (43), 139 (40), 105 (13), 91 (14). Calcd. C_44_H_54_N_2_O_5_, m/z [M]^+^ 690.4027. Found, *m*/*z*: 690.4042.

**Compound 18**. (4*S*,5′*R*,6a*R*,6b*S*,8a*S*,8b*R*,9*S*,10*R*,11a*S*,12a*S*,12b*S*)-5′,6a,8a,9-Tetra- methyl-1,3,3′,4,4′,5,5′,6,6a,6b,6′,7,8,8a,8b,9,11a,12,12a,12b-icosahydrospiro(naphtho[2′,1′: 4,5]indeno[2,1-b]furan-10,2′-pyran)-4-yl (*E*)-4-(4-fluorophenyl)-2-(2-phenylhydrazineyli- dene)but-3-ynoate [(22*R*,25*R*)-spirost-5-en-3β-yl (*E*)-4-(4-fluorophenyl)-2-(2-phenylhyd- razineylidene)but-3-ynoate] (**18**). Yield 27% (b). Yellow solid. M.p. 109 °C (decomp.). [α]_D_^24^—46.0 (c 0.2, CHCl_3_). ^1^H NMR (400 MHz, CDCl_3_, δ, ppm): 0.78 (3H, s, H-19), 0.77 (3H, d, *J* = 6.8 Hz, H-27),0.98 (3H, d, *J* = 7.0 Hz, H-C^21^H_3_), 1.02 (1H, m, H-9), 1.10 (3H, s, H-18), 1.13–1.35 (4H, m, H-1,12,14,15), 1.44–1.69 (9H, m, H-7,8,11,11,23,23,24,24,25), 1.72–1.82 (3H, m, H-2,12,17), 1.85–1.94 (2H, m, H-1,20), 1.98–2.06 (3H, m, H-2,7,15), 2.50 (2H, m, H-4), 3.39 (1H, t, J = 10.8 Hz, H-26), 3.48 (1H, dm, *J* = 10.8 Hz, H-26), 4.42 (1H, dd, *J* = 15.0, 7.5 Hz, H-16), 4.82 (1H, m, H-3), 5.42 (1H, d, *J* = 4.6 Hz, H-6), 7.08 (1H, d, *J* = 7.4 Hz, H-4‴), 7.12 (2H, t, *J* = 8.6 Hz, H-3″,5″), 7.29 (2H, d, *J* =7.9, H-2‴,6‴), 7.35 (2H, dd, *J* = 7.9, 7.3 Hz, H-3‴,5‴), 7.59 (2H, dd, *J* = 8.9, 5.4 Hz, H-2″,6″), 9.05 (1H, s, NH). ^13^C NMR (101 MHz, CDCl_3_, δ, ppm): 14.5 (C-21), 16.3 (C-18), 17.1 (C-27), 19.4 (C-19), 20.8 (C-11), 27.7 (C-2), 28.7 (C-24), 30.3 (C-25), 31.3 (C-23), 31.4 (C-8), 31.8 (C-15), 32.0 (C-7), 36.8 (C-10), 36.9 (C-1), 38.0 (C-4), 39.7 (C-12), 40.2 (C-13), 41.6 (C-20), 49.9 (C-9), 56.4 (C-14), 62.0 (C-17), 66.8 (C-26), 75.4 (C-3), 77.5 (C-3′), 80.8 (C-16), 103.6 (C-4′), 109.3 C-22), 114.8 (C-2‴,6‴), 115.9, 116.2 (C-2″,6″), 117.5 (C-1″), 117.6 (C-2′), 122.5 (C-6), 123.3 (C-4‴), 129.4 (C-3‴,5‴), 133.9, 134.0 (C-3″,5″), 139.7 (C-5), 141.7 (C-1‴), 162.6 (C-4″, J_C-F_ = 255.2 Hz), 162.3 (C-1′). IR (KBr, ν, cm^−1^): 3280 (NH), 2195 (C≡C), 1711 (C=O), 1529 (C=N), 1336, 1271, 1191, 1155 (C-F), 1068, 1051, 1026, 1007 (C-O-C), 1603, 1508, 816, 798, 752, 708, 690 (C=C). HR-MS, *m*/*z* (I_rel._, %): 678 (4), 397 (23), 396 (57), 395 (15), 394 (26), 283 (18), 282 (62), 211 (27), 139 (100), 115 (16). Calcd. C_43_H_51_FN_2_O_4_, *m*/*z* [M]^+^ 678.3827. Found, *m*/*z*: 678.3830.

**Compound 19**. (4*S*,5′*R*,6a*R*,6b*S*,8aS,8b*R*,9S,10*R*,11a*S*,12aS,12b*S*)-5′,6a,8a,9-Tetra- methyl-1,3,3′,4,4′,5,5′,6,6a,6b,6′,7,8,8a,8b,9,11a,12,12a,12b-icosahydrospiro[naphtho[2′,1′: 4,5]indeno[2,1-*b*]furan-10,2′-pyran]-4-yl (*E*)-2-(2-phenylhydrazineylidene)-4-(*m*-tolyl)- but-3-ynoate [(22*R*,25*R*)-spirost-5-en-3β-yl (*E*)-2-(2-phenylhydrazineyli- dene)-4-(*m*-tolyl)but-3-ynoate] (**19**). Yield 37% (b). Yellow solid. M.p. 93 °C (decomp.). [α]_D_^25^ -64.6 (c 0.2, CHCl_3_). ^1^H NMR (500 MHz, CDCl_3_, δ, ppm): 0.78 (3H, s, H-19), 0.77 (3H, d, *J* = 6.7 Hz, H-27), 0.98 (3H, d, *J* = 7.0, H-21), 1.02 (1H, m, H-9), 1.10 (3H, s, H-18), 1.14 (1H, m, H-14), 1.21 (2H, m, H-1,12), 1.30 (1H, m, H-15), 1.48 (2H, m, H-11,24), 1.55–1.72 (7H, m, H-7,8,11,23,23,24,25), 1.75–1.81 (3H, m, H-2,12,17), 1.89 (1H, q, *J* = 7.0 Hz, H-20), 1.92 (1H, dt, *J* = 13.6, 3.7 Hz, H-1), 1.99–2.06 (3H, m, H-2,7,15), 2.40 (3H, s, H-7″), 2.52 (2H, m, H-4), 3.39 (1H, t, *J* = 10.8 Hz, H-26), 3.48 (1H, dm, *J* = 10.8 Hz, H-26), 4.43 (1H, dd, *J* = 15.0, 7.3 Hz, H-16), 4.83 (1H, m, H-3), 5.42 (1H, d, *J* = 5.1 Hz, H-6), 7.05 (1H, t, *J* = 7.4 Hz, H-4‴), 7.25 (1H, d, *J* = 7.6 Hz, H-4″), 7.29–7.36 (5H, m, H-5″,2‴,6‴,3‴,5‴), 7.40 (1H, d, *J* = 7.9 Hz, H-6″), 7.42 (1H, s, H-2″), 9.09 (1H, s, NH). ^13^C NMR (126 MHz, CDCl_3_, δ, ppm): 14.5 (C-21), 16.3 (C-18), 17.1 (C-27), 19.4 (C-19), 20.8 (C-11), 21.2 (C-7″), 27.7 (C-2), 28.8 (C-24), 30.3 (C-25), 31.3 (C-23), 31.4 (C-8), 31.8 (C-15), 32.2 (C-7), 36.8 (C-10), 37.0 (C-1), 38.1 (C-4), 39.7 (C-12), 40.4 (C-13), 41.6 (C-20), 49.9 (C-9), 56.4 (C-14), 62.1 (C-17), 66.8 (C-26), 75.3 C-3), 77.4 (C-3′), 80.8 (C-16), 105.1 (C-4′), 109.2 (C-22), 114.8 (C-2‴,6‴), 117.9 (C-2′), 121.2 (C-3″), 122.4 (C-6), 123.2 (C-4‴), 128.5 (C-4″), 129.0 (C-5″), 129.4 (C-3‴,5‴), 130.6 (C-2″), 132.4 (C-6″), 138.4 (C-1″), 139.8 (C-5), 141.9 (C-1‴), 162.4 (C-1′). IR (KBr, ν, cm^−1^): 3281 (NH), 2189 (C≡C), 1711 (C=O), 1527 (C=N), 1068, 1051, 1026, 1007 (C-O-C), 1603, 1502, 835, 814, 783, 752, 688 (C=C). HR-MS, *m*/*z* (I_rel._, %): 674 (0.4), 397 (28), 396 (54), 283 (20), 282 (74), 143 (75), 139 (100), 105 (23), 91 (26), 69 (23). Calcd. C_44_H_54_N_2_O_4_. *m*/*z* [M]^+^ 674.4078. Found, *m*/*z*: 674.4073.

**Compound 20**. (4*S*,5′*R*,6a*R*,6b*S*,8a*S*,8b*R*,9*S*,10*R*,11a*S*,12a*S*,12b*S*)-5′,6a,8a,9-Tetra- methyl-1,3,3′,4,4′,5,5′,6,6a,6b,6′,7,8,8a,8b,9,11a,12,12a,12b-icosahydrospiro(naphtho[2′,1′: 4,5]indeno[2,1-*b*]furan-10,2′-pyran)-4-yl (*Z*)-4-phenyl-2-(2-phenylhydrazineylidene)but- 3-ynoate [(22*R*,25*R*)-spirost-5-en-3β-yl (*Z*)-4-phenyl-2-(2-phenylhydrazineylidene)but- 3-ynoate] (**20**). Yield 7% (c). Yellow crystals. M.p. 146 °C (decomp.). [α]_D_^25^ -93.8 (c 0.2, CHCl_3_). ^1^H NMR (300 MHz, CDCl_3_, δ, ppm): 0.77 (3H, s, H-19), 0.76 (3H, d, *J* = 6.8 Hz, H-27), 0.97 (3H, d, *J* = 6.8 Hz, H-21), 1.01 (1H, m, H-9), 1.07 (3H, s, H-18), 1.11–1.34 (4H, m, H-1,12,14,15), 1.41–2.02 (17H, m, H-1,2,2,7,7,8,11,11,12,15, 17,20,23,23,24, 24,25), 2.47 (2H, m, H-4), 3.36 (1H, t, *J* = 10.7 Hz, H-26), 3.45 (1H, dd, *J* = 10.7, 3.4 Hz, H-26), 4.39 (1H, dd, *J* = 14.8, 7.5 Hz, H-16), 4.72 (1H, m, H-3), 5.43 (1H, d, *J* = 4.1 Hz, H-6), 7.03 (1H, t, *J* = 6.7 Hz, H-4″), 7.24–7.34 (7H, m, H-2″,6″,2‴,4‴,3‴,5‴,6‴), 7.50–7.53 (2H, m, H-3″,5″), 12.8 (1H, s, NH). ^13^C NMR (126 MHz, CDCl_3_, δ, ppm: 14.5 (C-21), 16.3 (C-18), 17.1 (C-27), 19.4 (C-19), 20.8 (C-11), 27.6 (C-2), 28.8 (C-24), 30.3 (C-25), 31.3 (C-23), 31.4 (C-8), 31.8 (C-15), 32.0 (C-7), 36.8 (C-10), 36.9 (C-1), 37.9 (C-4), 39.7 (C-12), 40.2 (C-13), 41.6 (C-20), 49.9 (C-9), 56.4 (C-14), 61.9 (C-17), 66.8 (C-26), 75.6 (C-3), 80.8 (C-16) 85.3 (C-3′), 89.9 (C-4′), 109.3 (C-22), 113.9 (C-2′), 114.7 (C-4″), 122.8 (C-6), 123.0 (C-1″), 123.5 (C-4‴), 128.3 (2C), 128.4, 129.3 (2C), 131.5 (2C), 131.9 (C-2″,6″,3″,5″,2‴,6‴,3‴,5‴), 139.4 (C-5), 142.1 (C-1‴), 163.4 (C-1′). IR (KBr, ν, cm^−1^): 3325 (NH), 2192 (C≡C), 1715 (C=O), 1522 (C=N), 1068, 1051, 1023, 1008 (C-O-C), 1602, 1502, 838, 799, 787, 753, 688 (C=C). HR-MS, *m*/*z* (I_rel._, %): 660 (10), 397 (31), 396 (23), 324 (36), 282 (45), 264 (47), 193 (35), 139 (100), 115 (32), 93 (26), 91 (36), 77 (28). Calcd. C_43_H_52_N_2_O_4_, *m*/*z* [M]^+^ 660.3922. Found, 660.3918. X-ray structural analysis of compound **20**: C_43_H_52_N_2_O_4_, M 660.87, monoclinic, C2, a 31.72(1), b 7.361(2), c 18.279(5) Å, β 118.52(1)º, V 3750(2) Å^3^, Z 4, D_calcd_ 1.171 g·cm^−3^, μ(Mo-Kα) 0.074 mm^−1^, F(000) 1424, (θ 2.27–25.53°, completeness 100%), yellow, (0.26 × 0.21 × 0.04) mm^3^, transmission 0.7563–0.8596. The intensities of 6672 independent reflections were measured (R_int_ 0.1263), 446 parameters, 41 restraints, R_1_ 0.0554 (for 3766 observed I > 2σ(I)), wR_2_ = 0.1614 (all data), GOOF 1.024, largest diff. peak and hole 0.345 and -0.176 e.A^−3^. Crystallographic data for structure **20** have been deposited at the Cambridge Crystallographic Data Centre as supplementary publication no. CCDC 2077785.

**Compound 21**. (4*S*,5′*R*,6a*R*,6b*S*,8aS,8b*R*,9*S*,10*R*,11a*S*,12a*S*,12b*S*)-5′,6a,8a,9-Tetra- methyl-1,3,3′,4,4′,5,5′,6,6a,6b,6′,7,8,8a,8b,9,11a,12,12a,12b-icosahydrospiro(naphtho[2′,1′: 4,5]indeno[2,1-*b*]furan-10,2′-pyran)-4-yl 1,3-diphenyl-1*H*-pyrazole-5-carboxylate [(22*R*,25*R*)-spirost-5-en-3β-yl 1,3-diphenyl-1*H*-pyrazole-5-carboxylate] (**21**). Yield 7% (c). Yellow needles. M.p. 96 °C (decomp.) (petroleum ether–ether, 20:1). [α]_D_^24^–76.1 (c 0.2, CHCl_3_). ^1^H NMR (600 MHz, CDCl_3_, δ, ppm): 0.77 (3H, s, H-19), 0.78 (3H, d, *J* = 6.9 Hz, H-27), 0.97 (3H, d, *J* = 6.9 Hz, H-21), 1.03 (3H, s, H-18), 1.06–1.33, 1.45–1.69, 1.73–1.80, 1.84–1.89, 1.96–2.02 (21H, m, H-1,1,2,2,4,4,7,7,8,9,11,11,12,12,14,15,15,20,24,24,25), 2.27 (1H, t, *J* = 12.1 Hz, H-23), 2.33 (1H, dt, *J* = 12.1, 3.4 Hz, H-23), 3.38 (1H, t, *J* = 10.9 Hz, H-26), 3.48 (1H, dm, *J* = 10.9 Hz, H-26), 4.41 (1H, dd, *J* = 15.2, 7.7 Hz, H-16), 4.72 (1H, m, H-3), 5.37 (1H, d, *J* = 4.7 Hz, H-6). 7.34 (1H, s, H-4′), 7.36, 7.43, 7.47–7.49, 7.88 (10H, m, H-2″,3″,4″,5″,6″,2‴,3‴,4‴,5‴,6‴). ^13^C NMR (126 MHz, CDCl_3_, δ, ppm): 14.5 (C-21), 16.2 (C-18), 17.1 (C-27), 19.2 (C-19), 20.8 (C-11), 27.5 (C-2), 28.8 (C-24), 30.3 (C-25), 31.3 (C-8), 31.4 (C-23), 31.8 (C-15), 32.0 (C-7), 36.7 (C-10), 36.83 (C-1), 37.8 (C-4), 39.7 (C-12), 40.2 (C-13), 41.6 (C-20), 49.9 (C-9), 56.4 (C-14), 62.1 (C-17), 66.8 (C-26), 75.0 (C-3), 80.8 (C-16), 109.2 (C-22), 109.4 (C-4′), 122.7 (C-6), 125.8 (2C), 126.2 (2C), 128.3, 128.5, 128.6 (2C), 128.7 (2c) (C-2″,3″,4″,5″,6″, 2‴,3‴,4‴,5‴,6‴), 132.2 (C-1″), 135.1 (C-3′), 139.3 (C-1‴), 140.5 (C-5), 151.4 (C-5′), 158.5 (C=O). IR (KBr, ν, cm^−1^): 1728 (C=O), 1543 (C=N pyrazole), 982, 1072, 1053, 1009 (C-O-C), 1599, 1500, 833, 796, 762, 692 (C=C). HR-MS, *m*/*z* (I_rel._, %): 660 (2), 397 (33), 396 (100.00), 283 (26), 282 (98), 265 (87), 264 (55), 247 (25), 139 (95), 69 (29). Calcd. C_43_H_52_N_2_O_4_, *m*/*z* [M]^+^ 660.3922. Found 660.3897. X-ray structural analysis of compound **21**: C_43_H_52_N_2_O_4_, M 660.87, monoclinic, P2_1_, a 14.2635(6), b 9.7194(4), c 26.781(1), Å, β 95.085(2)°, V 3698.1(3) Å^3^, Z 4, D_calcd_ 1.187.0g·cm^−3^, μ(Mo-Kα) 0.075 mm^−1^, F(000) 1424, (θ 0.76–25.06°, completeness 99.9%), colorless, (1.0 × 0.14 × 0.03) mm^3^, transmission 0. 0.7472–0.8620. The intensities of 12972 independent reflections were measured (R_int_ 0.0587), 891 parameters, 4 restraints, R_1_ 0.0499 (for 8000 observed I > 2σ(I)), wR_2_ = 0.1360 (all data), GOOF 1.058, largest diff. peak and hole 0.174 and -0.214 e.A^−3^. Crystallographic data for structure **21** have been deposited at the Cambridge Crystallographic Data Centre as supplementary publication no. CCDC 2077786.

##### Four Component Reaction of Diosgenin **1**, Oxalyl Chloride, Phenyl Acetylene **3a** and Hydrazine Monohydrate (**22**)

A solution of diosgenin **1** (1.50 g, 3.6 mmol) in chloroform (20 mL) was added dropwise to a cold stirred solution of oxalyl chloride (1.24 mL, 14.5 mmol) in chloroform (3 mL) for an hour under argon flow. The reaction mixture was warmed to room temperature and stirred for 3 h. The solvent was evaporated, the residue was diluted with chloroform (3 mL) and additionally evaporated. This procedure was repeated three times for removing the trace of oxalyl chloride. The crude residue of 3-*O*-(2-chloro-2-oxoacetyl)-diosgenin **2** was dissolved in benzene (15 mL) and phenylacetylene **3a** (0.37 g, 3.6 mmol), CuI (0.069 g, 0.36 mmol) and Et_3_N (0.50 mL, 3.6 mmol) were added under stirring in an argon flow. The reaction mixture was warmed to 40 °C and stirred for 12 h (TLC). The solvent was removed under reduced pressure, and 2-methoxyethanol (15 mL) was added. The solution was treated with hydrazine hydrate **22** (0.217 g, 4.3 mmol) and stirred in the argon flow for 24 h at ambient temperature (TLC). After evaporation the residue was purified by column chromatography (petroleum ether–ethylacetate, 5:1) to afford 0.967 g (46%) of compound **23**.

**Compound 23**. (4*S*,5′*R*,6a*R*,6b*S*,8a*S*,8b*R*,9*S*,10*R*,11a*S*,12a*S*,12b*S*)-5′,6a,8a,9-Tetra- methyl-1,3,3′,4,4′,5,5′,6,6a,6b,6′,7,8,8a,8b,9,11a,12,12a,12b-icosahydrospiro(naphtho[2′,1′: 4,5]indeno[2,1-*b*]furan-10,2′-pyran)-4-yl 3-phenyl-1*H*-pyrazole-5-carboxylate [(22*R*,25*R*)-spirost-5-en-3β-yl 3-phenyl-1*H*-pyrazole-5-carboxylate] (**23**). Yield 46%. White needles. M.p. 274–276 °C. [α]_D_^26^—63.2 (c 0.14, CHCl_3_). ^1^H NMR (600 MHz, CDCl_3_, δ, ppm): 0.75 (3H, s, H-19), 0.78 (3H, d, *J* = 6.8 Hz, H-27), 0.98 (3H, d, *J* = 7.0 Hz, H-21), 1.02 (1H, m, H-9), 1.05 (3H, s, H-18), 1.13 (1H, m, H-14), 1.21 (2H, m, H-1,12), 1.31 (1H, m, H-15), 1.45–1.56 (3H, m, H-11,11,24), 1.58–1.71 (6H, m, H-7,8,20,23,23,24), 1.74 (1H, dm, *J* = 12.8, H-12), 1.79 (1H, dd, *J* = 8.4, 6.6 Hz, H-17), 1.88 (2H, m, H-1,25), 1.99–2.05 (3H, m, H-2,7,15), 2.44 (2H, m, H-4), 3.39 (1H, t, *J* = 10.8 Hz, H-26), 3.48 (1H, dd, *J* = 10.8, 3.6 Hz, H-26), 4.42 (1H, dd, *J* = 15.4, 7.7 Hz, H-16), 4.86 (1H, m, H-3), 5.41 (1H, d, *J* = 4.4 Hz, H-6), 7.10 (1H, s, H-4′), 7.36 (1H, t, *J* = 7.3 Hz, H-4″), 7.43 (2H, dd, *J* = 7.7, 7.3 Hz, H-3″,5″), 7.7 (2H, d, *J* = 7.7 Hz, H-2″,6″). ^13^C NMR (151 MHz, CDCl_3_) δ_C_: 14.5 (C-21), 16.3 (C-18), 17.1 (C-27), 19.3 (C-19), 20.8 (C-11), 27.7 (C-2), 28.8 (C-24), 30.3 (C-25), 31.3 (C-23), 31.4 (C-8), 31.8 (C-15), 32.1 (C-7), 36.7 (C-10), 36.9 (C-1), 38.0 (C-4), 39.7 (C-12), 40.2 (C-13), 41.6 (C-20), 49.9 (C-9), 56.4 (C-14), 62.1 (C-17), 66.8 (C-26), 75.1 (C-3), 80.8 (C-16), 105.5 C-4′), 109.3 (C-22), 122.7 (C-6), 125.9 (C-2″,6″), 128.5 (C-4″), 128.9 (C-3″,5″), 131.1 (C-1″), 138.5 (C-5′), 139.4 (C-5), 150.3 (C-3′), 159.9 (C=O). IR (KBr, ν, cm^−1^): 3369 (NH), 1709 (C=O), 1456, 1554 (C=N pyrazole), 1070, 1051, 1026, 1009 (C-O-C), 1612, 1514, 843, 827, 777, 766, 721, 692 (C=C). Found, %: C 76.32; H 8.31; N 5.05. Calcd. for C_37_H_48_N_2_O_4_: C 75.99; H 8.27; N 4.79. X-ray structural analysis of compound **23**: C_37_H_48_N_2_O_4_, M 584.77, monoclinic, P2_1_, a 16.8479(7), b 6.0052(2), c 16.9035(8) Å, β 108.231(2)º, V 1624.4(1) Å^3^, Z 2, D_calcd_ 1.196g·cm^−3^, μ(Mo-Kα) 0.077 mm^−1^, F(000) 632, (θ 1.27–25.07°, completeness 99.8%), yellow, (1.0 × 0.20 × 0.04) mm^3^, transmission 0. 7988–0.8620. The intensities of 5753 independent reflections were measured (R_int_ 0.0382), 392 parameters, 21 restraints, R_1_ 0.0512 (for 4476 observed I > 2σ(I)), wR_2_ = 0.1550 (all data), GOOF 1.093, largest diff. peak and hole 0.333 and -0.182 e.A^−3^. Crystallographic data for structure **23** have been deposited at the Cambridge Crystallographic Data Centre as supplementary publication no. CCDC 2077787.

##### Four Component Reaction of Diosgenin 1, Oxalyl Chloride, Phenyl Acetylene **3a** and N-Acyl Substituted Hydrazines **24a,b**

A solution of diosgenin **1** (0.80 g, 1.93 mmol) in chloroform (13 mL) was added was added dropwise to a cold stirred solution of oxalyl chloride (0.66 mL, 7.96 mmol) (0 °C) in chloroform (3 mL) under argon flow. The reaction mixture was warmed to ambient temperature and stirred for 3 h. The solvent was removed under reduced pressure, the residue was diluted with chloroform (3 mL) and additionally evaporated. This procedure was repeated three times for removing the trace of oxalyl chloride. The crude residue of **2** was dissolved in benzene (13 mL) and phenylacetylene **3a** (0.20 g, 1.93 mmol), CuI (0.037 g, 0.19 mmol) and Et_3_N (0.30 mL, 2.17 mmol) were added under stirring in an argon flow. The reaction mixture was warmed to 40 °C and stirred for 12 h (TLC), then the N-acyl hydrazine **24a** or **24b** (1.61 mmol) was added. After stirring at 40 °C for 24 h (TLC) the solvent was removed under reduces pressure and the residue was purified by column chromatography for isolation of compounds **25**, **26** (eluent petroleum ether–ethylacetate, 100:15, and petroleum ether–ethylacetate, 5:1) for isolation of pyrazole **23**.

**Compound 25**. (4*S*,5′*R*,6a*R*,6b*S*,8a*S*,8b*R*,9*S*,10*R*,11a*S*,12a*S*,12b*S*)-5′,6a,8a,9-Tetra- methyl-1,3,3′,4,4′,5,5′,6,6a,6b,6′,7,8,8a,8b,9,11a,12,12a,12b-icosahydrospiro(naphtho[2′,1′: 4,5]indeno[2,1-b]furan-10,2′-pyran)-4-yl 2-benzoyl-3-hydroxy-5-phenyl-2,3-dihydro-1*H*- pyrazole-3-carboxylate [(22*R*,25*R*)- spirost-5-en-3β-yl 2-benzoyl-3-hydroxy-5-phenyl- 2,3-dihydro-1*H*-pyrazole- 3-carboxylate] (**25**). Yield 49%. White solid. [α]_D_^26^ -53.9 (c 0.3, CHCl_3_). ^1^H NMR (600 MHz, CDCl_3_, δ, ppm): 0.78 (3H, s, H-19), 0.79 (3H, d, *J* = 6.6 Hz, H-27), 0.97 (3H, d, *J* = 7.0 Hz, H-21), 1.02 (3H, s, H-18), 1.00 (1H, m, H-9), 1.08–1.22 (3H, m, H-1,12,14), 1.27 (1H, m, H-15), 1.40–1.69 (9H, m, H-7,8,11,11,23,23,24,24,25), 1.71–1.80 (3H, m, H-2,12,17), 1.83–2.02 (5H, m, H-1,2,7,15,20), 2.27–2.42 (2H, m, H-4), 3.38 (1H, t, *J* = 10.9 Hz, H-26), 3.43 (1H, d, *J* = 17.9 Hz, H-4′), 3.49 (1H, m, H-26), 3.60 (1H, d, *J* = 17.9 Hz, H-4′) 4.41 (1H, dd, *J* = 15.0, 7.7 Hz, H-16), 4.82 (1H, m, H-3), 5.05 (1H, s, H-OH), 5.40 (1H, s, H-6), 7.40–7.49 (5H, m, H-3″,4″,5″,3‴,5‴), 7.55 (1H, t, *J* = 7.4 Hz, H-4‴), 7.70 (2H, d, *J* = 8.2 Hz, H-2″,6″), 8.03 (2H, dd, *J* = 7.4, 8.2 Hz, H-2‴,6‴). ^13^C NMR (151 MHz, CDCl_3_, δ, ppm): 14.5 (C-21), 16.2 (C-18), 17.1 (C-27), 19.3 (C-19), 20.8 (C-11), 27.4, 27.53 (C-2), 28.79 (C-24), 30.27 (C-25), 31.35 (C-8), 31.37 (C-23), 31.81 (C-15), 32.0 (C-7), 36.6 (C-10), 36.7, 36.8 (C-1), 37.6, 37.8 (C-4), 39.7 (C-12), 40.2 (C-13), 41.6 (C-20), 44.8 (C-4′), 49.9 (C-9), 56.4 (C-14), 62.1 (C-17), 66.8 (C-26), 76.9 (C-3), 80.8 (C-16), 89.5, 89.6 (C-5′), 109.2 (C-22), 122.8, 122.9 (C-6), 126.7, 127.8, 130.2, 130.6, 131.6 (all 2C-2″,3″,4″,5″,6″,2‴, 3‴, 4‴,5‴,6‴), 130.8 (C-1″), 132.9 (C-1‴), 139.0, 139.2 (C-5), 152.8 (C-3′), 166.9 (PhC=O), 169.4 (C=O). IR (KBr, ν, cm^−1^): 3458 (OH), 1730, 1687 (C=O), 1452, 1572 (C=N), 1078, 1055, 1032, 1007 (C-O-C), 1620, 1500, 877, 787, 756, 700 (C=C). HR-MS. Calcd. C_44_H_54_N_2_O_6_, *m*/*z* [M] ^+^ 706.40. Found = 707.405 [M+H]^+^.

**Compound 26**. (4*S*,5′*R*,6a*R*,6b*S*,8a*S*,8b*R*,9*S*,10*R*,11a*S*,12a*S*,12b*S*)-5′,6a,8a,9-Tetra- methyl-1,3,3′,4,4′,5,5′,6,6a,6b,6′,7,8,8a,8b,9,11a,12,12a,12b-icosahydrospiro[naphtho[2′,1′: 4,5]indeno[2,1-*b*]furan-10,2′-pyran]-4-yl 1-(4-bromobenzoyl)-5-hydroxy-3-phenyl-4,5- dihydro-1*H*-pyrazole-5-carboxylate [(22*R*,25*R*)-spirost-5-en-3β-yl 1-(4-bromobenzoyl)- 5-hydroxy-3-phenyl-4,5-dihydro-1*H*-pyrazole- 5-carboxylate] (**26**). Yield 42%. Grey solid. [α]_D_^2^^4^ -15.0 (c 0.4, CHCl_3_). ^1^H NMR (600 MHz, CDCl_3_, δ, ppm): 0.78 (3H, s, H-19), 0.79 (3H, d, *J* = 6.5 Hz, H-27), 0.97 (3H, d, *J* = 6.9 Hz, H-21), 1.03 (3H, s, H-18), 1.00 (1H, m, H-9), 1.07–1.22 (3H, m, H-1,12,14), 1.29 (1H, m, H-15), 1.40–1.69 (9H, m, H-7,8,11,11,23,23,24,24, 25), 1.71–1.80 (3H, m, H-2,12,17), 1.83–2.02 (5H, m, H-1,2,7,15,20), 2.23–2.39 (2H, m, H-4), 3.38 (1H, t, *J* = 11.0, H-26), 3.43 (1H, d, *J* = 17.9 Hz, H-4′), 3.49 (1H, m, H-26), 3.60 (1H, d, *J* = 17.9 Hz, H-4′) 4.41 д.д (1H, dd, *J* = 15.2, 7.7, H-16), 4.81 (1H, m, H-3), 5.00 (1H, s, H-OH), 5.40 (1H, s, H-6), 7.46 м (3H, m, H-3″,4″,5″), 7.61 (2H, d, *J* = 8.5, H-3‴,5‴), 7.69 (2H, d, *J* = 7.8, H-2″,6″), 7.92 (2H, d, *J* = 8.1, H-2‴,6‴). ^13^C NMR (151 MHz, CDCl_3_, δ, ppm): 14.5 (C-21), 16.2 (C-18), 17.1 (C-27), 19.3 (C-19), 20.8 (C-11), 27.4, 27.5 (C-2), 28.8 (C-24), 30.3 (C-25), 31.3 (C-8), 31.4 (C-23), 31.8 (C-15), 32.0 (C-7), 36.6 (C-10), 36.7, 36.8 (C-1), 37.6, 37.8 (C-4) 39.6 (C-12), 40.2 (C-13), 41.6 (C-20), 44.8 (C-4′), 49.8 (C-9), 56.4 (C-14), 62.0 (C-17), 66.8 (C-26), 76.8 (C-3), 80.8 (C-16), 89.5, 89.6 (C-5′), 109.2 (C-22), 122.9, 123.0 (C-6), 126.5 (C-1‴), 126.8 (C-2″,6″), 128.8 (C-3″,5″), 130.5 (C-1″), 130.8 (C-3‴,5‴), 131.1 (C-2‴,6‴), 131.7 (C-4‴), 131.8 (C-4″), 138.93, 139.05 (C-5), 153.2 (C-3′), 165.8 (ArC=O), 169.2 (C=O). IR (KBr, ν, cm^−1^): 3458 (OH), 1730, 1689 (C=O), 1467, 1545 (C=N), 1072, 1053, 1011, 996 (C-O-C), 1598, 1500, 837, 823, 758, 746, 690 (C=C). Found, %: C 65.37; H 6.69; N 2.98; Br 9.03. Calcd. for C_44_H_53_N_2_O_6_Br: C 67.25; H 6.80; N 3.36; Br 9.47.

### 3.2. Biological Studies

#### 3.2.1. Animals

The pharmacological studies were carried out on CD-1 mice and C57BL/6 female mice weighing 20–25 g with eight animals in each group. All animals were taken from the SPF-vivarium of the ICG SB RAS (Institute of Cytology and Genetics of Russian Academy of Sciences Siberian Branch). Mice were housed in wire cages at 22–25 °C on a 12 h light–dark cycle. The animals had free access to a standard pellet diet, and tap water was available ad libitum. All experimental procedures were approved by the Bio-Ethical Committee of Medicine Chemistry Department of Novosibirsk Institute of Organic Chemistry SB RAS (Protocol Code P-1-03.2020-14) in accordance with the European Convention for the Protection of Vertebrate Animals used for Experimental and other Scientific Purposes, and the requirements and recommendations of the Guide for the Care and Use of Laboratory Animals.

#### 3.2.2. Anti-Inflammatory Activity

The anti-inflammatory activity of compounds 1,7,8,10,13–16,18,19,21,23 and 26 was evaluated using the histamine-induced mouse paw edema models. The test compounds were dissolved in distilled water containing 0.5% Tween 80 just before intraperitoneal administration at the dose of 50 mg/kg. The reference drug diclofenac sodium (‘Fluka BioChemica’) was administered by the same method at the effective dose 10 mg/kg. The control group of animals received an equivalent volume of water–Tween mixture. All the mice were subplantarly injected into the hind paw with 0.05 mL 0.01% histamine (“Sigma Aldrich”, Saint Louis, MO, USA) in a water solution. The tested compounds were administered 1 h before the histamine injection. The animals were sacrificed by cervical dislocation 5 h after the phlogogen injection. Inflammation percentage was calculated by the equation: %inflammation = 100× [(Mip − Mnip)/Mnip], where Mip is the inflamed paw mass and Mnip the non-inflamed paw mass [54].

Statistical analysis data were presented as mean ± standard error (SEM) from groups of animals. Statistical analysis was applied with parametric and non-parametric methods using “STATISTICA 6” software. The differences were considered significant at *p* < 0.05.

### 3.3. Molecular Docking Procedure

Molecular modeling was carried out in the Schrodinger Maestro visualization environment using applications from the *Schrodinger Small Molecule Drug Discovery Suite 2016-1* package [61]. Three-dimensional structures of the derivatives were obtained empirically in the LigPrep application using the OPLS3 force field [62]. All possible tautomeric forms of compounds, as well as various states of polar protons of molecules in the pH range of 7.0 ± 2.0, were taken into account. For the calculations, the XRD model of the Kelch domain of the Keap1 protein with PDB ID 4IQK [63] (resolution 1.97 Å) was chosen. To model a possible mechanism of inhibition of the selected target, molecular docking of new compounds was performed at the binding site of the Cpd16 in the *Glide* application [64]. The search area for docking was selected automatically, based on the size and physio-chemical properties of the Cpd16 inhibitor. The extra precision (XP) algorithm of docking was applied. Docking was performed in comparison with Cpd16 inhibitor. The three-dimensional structures of inhibitors were obtained in the PubChem database and prepared in the *LigPrep* application. Non-covalent interactions of compounds in the binding site were visualized using *Biovia Discovery Studio Client* [65].

## 4. Conclusions

We report the sequential one-pot synthesis, including the steps of acylation of the alcohol substrate diosgenin with oxalyl chloride, the Stephens–Castro reaction of the formed 3-*O*-(2-chloro-2-oxoacetyl)diosgenin with arylalkynes and selective heterocyclization of alkynyldiones with hydrazines and hydrazides. We found the conditions of selective one-pot formation of 5-aryl substituted 3-*O*-(pyrazol-3-yloxo)diosgenin derivatives. By using 2-methoxyethanol at the step of reaction of spirostene alkyne-1,2-diones with phenylhydrazine hydrochloride regioisomeric α,β-alkynic hydrazones were also isolated. Carrying out the heterocyclization step of ynedione with hydrazine monohydrate in 2-methoxyethanol allowed the synthesis of 5-phenyl substituted steroidal pyrazole, which was found to exhibit high anti-inflammatory activity, comparable to that of diclofenac sodium. By in silico experiments, it was shown that the obtained steroidal α,β-alkynyl (*E*)-hydrazones and a diosgenin-pyrazole conjugate are incorporated into the binding site of the protein Keap1 Kelch-domain by their substituent at the hydroxyl group and form more non-covalent bonds, and have higher affinity than the initial spirostene core. It is foreseeable that the described one-pot consecutive process will find broad application in drug discovery and other related research fields.

## Data Availability

Data regarding synthesis, isolation and characterization are available upon request from M.E.M. Information related to the biological activity studies is available from S.A.B.

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
