# Peer review of "Synthesis of Anti-Inflammatory Spirostene-Pyrazole Conjugates by a Consecutive Multicomponent Reaction of Diosgenin with Oxalyl Chloride, Arylalkynes and Hydrazines or Hydrazones"

_molecules, 2021, doi:10.3390/molecules27010162_

Round 1

Reviewer 1 Report

Enjoyed reading this. A few minor suggestions.

In the paper:

Line 22- change to "supported by NMR spectroscopy" You may want to mention that you characterized all these compounds with mass spec.

Line 25-Define Diclofenac sodium, i.e., Diclofenac sodium, a commercial pain reliever

Line 44-practically insoluble sounds too informal

Line 63- cite use as antibacterial

Line 93-Did not led to sounds awkward

Line 112-Figure 1 caption separate page than the figure

Line 123- TLC control sounds strange, maybe something like as indicated by TLC

Line 130-Figure 2 caption not in line with text

Line 139- Don't just say "an analysis of the structure"... be more specific

Line 159- Don't start the paragraph with So

Scheme on line 170-what is 3kB after 1.2, should that just be equiv?

line 180- not properly formatted, i.e., figure 3 and figure 4's captions don't line up. Also, "The thermal ellipsoids are drawn at the 30% probability level"...should be included every time you include a crystal. This makes it more consistent.

Line 196 (scheme)-you used equivb... this should just be equiv

Line 200-201-This sentence..."The 1H and 13C NMR.... allow structure assignments " is not needed. This should be obvious

Line 215-217- I'm confused how only one doubling of resonances indicates a mixture of diastereomers. Shouldn't many 13C resonances be present if mixtures of diastereomers.

Line 284-Be more specific with "the diseases" what diseases are you talking about.

Line 365-participate is not the correct word-check this everywhere

Line 395 1H NMR not correctly formatted

Line 569-Bold both parts of the ( )

Line 861-added dropwise by cooling does not make sense.

Line 863- 3 hours should be abbreviated

Line 867- CuI(I) <-check for consistency, shouldn't this just be CuI

General suggestions manuscript

Also double check that all J's are italicized when presenting NMRs

I would list compound name first in bold followed by the name, i.e., Compound 5 (long IUPAC Name) this will make it easier for the reader to navigate your experimental.

Consider talking about synthesis, giving all of the synthetic characterization (e.g., experimental after you talk about synthesis), and then talk about bioactivity

Check to make sure 1H everywhere and not 1H.

SI-

In the NMR's you should try to integrate every peak. When you have a large grouping of peaks...i.e., 2.0 ppm - 1.0 ppm integrate the grouping so we can see that the hydrogens match up to the reported structure.

Author Response

Authors were very grateful for the valuable remarks from Referee 1. We have worked through all comments carefully and made the corresponding (many) changes to the manuscript. Please find below our answers in a point by point manner. We ask you to find the time and look at our additional comments.

  • Line 22 and line – 25 in the Abstract
  • The structure of new compounds was unambiguously corroborated by comprehensive NMR spectroscopy, mass-spectrometry and X-ray structure analyses. Carrying out the heterocyclization step of ynedione with hydrazine monohydrate in 2-methoxyethanol allowed the synthesis of 5-phenyl substituted steroidal pyrazole, which was found to exhibit high anti-inflammatory activity, comparable to that of Diclofenac sodium, a commercial pain reliever. the SAR is not informative. The main insights about their cytotoxicity have been directly linked to differences in the structures of the compounds and this is only a speculation, since the target is not know, but only hypothesized.
  • Line 44
  • diosgenin is poorly soluble in physiological media and has low absorption and a high percentage of the absorbed drug is metabolized rapidly

Line 63-

The Reference was added

Line 93- gave no acceptable results.

Line 112 Thank you very much

Line 123- Thank you very much

Line 130 - Thank you very much

Line 139-

Based on the analysis of the spectral and analytical data of obtained pyrazoles

Line 159- It is noteworthy that in mild reaction

Line 170, 180, 196 - - Thank you very much

Line 200-201-

  This was an error. The revised text:

The formation of a mixture of diastereomeric 5-hydroxypyrazolines is indicated by the doubling of some signals in the 13C NMR spectrum; the largest difference was observed for the chemical shifts of carbon atoms C-5ʹ (δ 89.5; 89.6 ppm) (Suppl.part).

Line 215-217.

This error was corrected.

Line 284- chronic diseases Line 395 1H NMR not correctly formatted

Line 569, Line 861, Line 863, Line 867- All this errors were corrected

Also double check that all J's are italicized when presenting NMRs

I would list compound name first in bold followed by the name, i.e., Compound 5 (long IUPAC Name) this will make it easier for the reader to navigate your experimental.

Thank you very much. This was added

SI-

We made some correction in the presentation of Spectra in SI.

We made important corrections and additions to the manuscript, which were necessary for the better presentation of our scientific material.

Thank you very much for all the comment.  

Reviewer 2 Report

the work is very good and well presented.

I would like to be defined in the discussion with any appropriate way, the drugability/druglikeness of these conjugates since in the introduction is referred that "diosgenin is
practically insoluble in physiological media and has low absorption ...., the aim of improving the biological activity and drugability properties...."

Author Response

Referee_2 Manuscript ID: molecules-1501631

Comments and Suggestions for Authors

the work is very good and well presented.

I would like to be defined in the discussion with any appropriate way, the drugability/druglikeness of these conjugates since in the introduction is referred that "diosgenin is
practically insoluble in physiological media and has low absorption ...., the aim of improving the biological activity and drugability properties...."

We appreciate of the helpful suggestions from Referee 2.

From the obtained to this time results we cannot defined the drugability/druglikeness of these conjugates. Therefore we made some change in the  Introduction part.

Thank you very mush for you consideration.

Round 2

Reviewer 2 Report

The authors have made satisfing changes. I do not have any comment. It can be accepted
Regards